# Instant3D: Fast Text-to-3D with Sparse-View Generation and Large Reconstruction Model

**Jiahao Li**[1,2*]   **Hao Tan**[1]   **Kai Zhang**[1]   **Zexiang Xu**[1]   **Fujun Luan**[1]   **Yinghao Xu**[1,3]
**Yicong Hong**[1,4]   **Kalyan Sunkavalli**[1]   **Greg Shakhnarovich**[2]   **Sai Bi**[1]

[1]Adobe Research   [2]TTIC   [3]Stanford University   [4] Australian National Univeristy

{jiahao,greg}@ttic.edu   yhxu@stanford.edu   mr.yiconghong@gmail.com
{hatan,kaiz,zexu,fluan,sunkaval,sbi}@adobe.com

## Abstract

Text-to-3D with diffusion models has achieved remarkable progress in recent years. However, existing methods either rely on score distillation-based optimization which suffer from slow inference, low diversity and Janus problems, or are feed-forward methods that generate low-quality results due to the scarcity of 3D training data. In this paper, we propose Instant3D, a novel method that generates high-quality and diverse 3D assets from text prompts in a feed-forward manner. We adopt a two-stage paradigm, which first generates a sparse set of four structured and consistent views from text in one shot with a fine-tuned 2D text-to-image diffusion model, and then directly regresses the NeRF from the generated images with a novel transformer-based sparse-view reconstructor. Through extensive experiments, we demonstrate that our method can generate diverse 3D assets of high visual quality within 20 seconds, which is two orders of magnitude faster than previous optimization-based methods that can take 1 to 10 hours. Our project webpage is: https://jiahao.ai/instant3d/.

## 1 Introduction

In recent years, remarkable progress has been achieved in the field of 2D image generation. This success can be attributed to two key factors: the development of novel generative models such as diffusion models (Song et al., 2021; Ho et al., 2020; Ramesh et al., 2022; Rombach et al., 2021), and the availability of large-scale datasets like Laion5B (Schuhmann et al., 2022). Transferring this success in 2D image generation to 3D presents challenges, mainly due to the scarcity of available 3D training data. While Laion5B has 5 billion text-image pairs, Objaverse-XL (Deitke et al., 2023a), the largest public 3D dataset, contains only 10 million 3D assets with less diversity and poorer annotations. As a result, previous attempts to directly train 3D diffusion models on existing 3D datasets (Luo & Hu, 2021; Nichol et al., 2022; Jun & Nichol, 2023; Gupta et al., 2023; Chen et al., 2023b) are limited in the visual (shape and appearance) quality, diversity and compositional complexity of the results they can produce.

To address this, another line of methods (Poole et al., 2022; Wang et al., 2023a; Lin et al., 2023; Wang et al., 2023b; Chen et al., 2023c) leverage the semantic understanding and high-quality generation capabilities of pretrained 2D diffusion models. Here, 2D generators are used to calculate gradients on rendered images, which are then used to optimize a 3D representation, usually a NeRF (Mildenhall et al., 2020). Although these methods yield better visual quality and text-3D alignment, they can be incredibly time-consuming, taking hours of optimization for each prompt. They also suffer from artifacts such as over-saturated colors and the "multi-face" problem arising from the bias in pretrained 2D diffusion models, and struggle to generate diverse results from the same text prompt, with varying the random seed leading to minor changes in geometry and texture.

In this paper, we propose Instant3D, a novel feed-forward method that generates high-quality and diverse 3D assets conditioned on the text prompt. Instant3D, like the methods noted above, builds on top of pretrained 2D diffusion models. However, it does so by splitting 3D generation into

---

*This work was done while the author was an intern at Adobe Research.

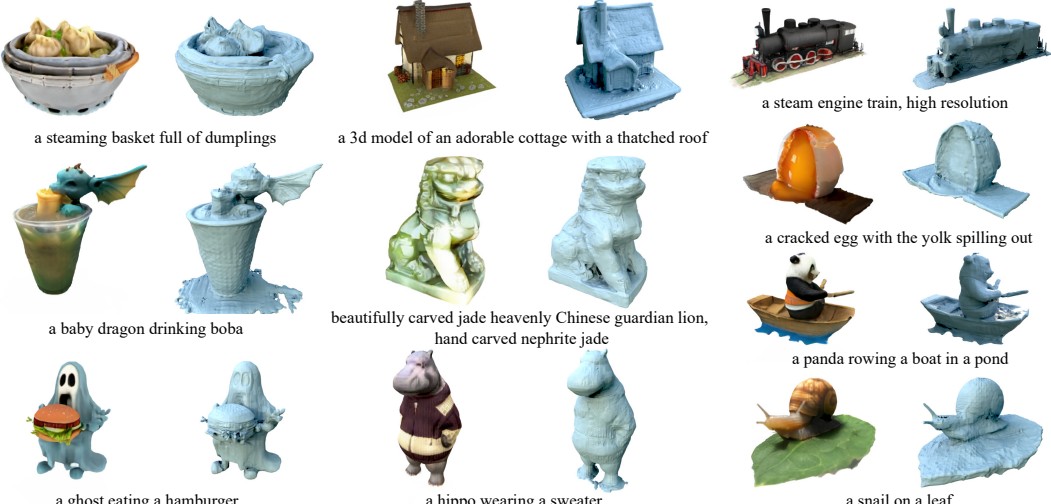

a steaming basket full of dumplings

a 3d model of an adorable cottage with a thatched roof

a steam engine train, high resolution

a baby dragon drinking boba

beautifully carved jade heavenly Chinese guardian lion, hand carved nephrite jade

a cracked egg with the yolk spilling out

a panda rowing a boat in a pond

a ghost eating a hamburger

a hippo wearing a sweater

a snail on a leaf

Figure 1: Our method generates high-quality 3D NeRF assets from the given text prompts within 20 seconds. Here we show novel view renderings from our generated NeRFs as well as the renderings of the extracted meshes from their density field.

two stages: 2D generation and 3D reconstruction. In the first stage, instead of generating images sequentially (Liu et al., 2023b), we fine-tune an existing text-to-image diffusion model (Podell et al., 2023) to generate a sparse set of four-view images in the form of a $2 \times 2$ grid in a single denoising process. This design allows the multi-view images to attend to each other during generation, leading to more view-consistent results. In the second stage, instead of relying on a slow optimization-based reconstruction method, inspired by Hong et al. (2024), we introduce a novel sparse-view *large reconstruction model* with a transformer-based architecture that can directly regress a triplane-based (Chan et al., 2022) NeRF from a sparse set of multi-view images. Our model projects sparse-view images into a set of pose-aware image tokens using pretrained vision transformers (Caron et al., 2021), which are then fed to an image-to-triplane decoder that contains a sequence of transformer blocks with cross-attention and self-attention layers. Our proposed model has a large capacity with more than 500 million parameters and can robustly infer correct geometry and appearance of objects from just four images.

Both of these stages are fine-tuned/trained with multi-view rendered images of around 750K 3D objects from Objaverse (Deitke et al., 2023b), where the second stage makes use of the full dataset and the first stage can be fine-tuned with as little as 10K data. While we use a relatively smaller dataset compared to the pre-training dataset for other modalities (e.g., C4 Raffel et al. (2020) for text and Laion5B for image), by combining it with the power of pretrained 2D diffusion models, Instant3D's two-stage approach is able to generate high-quality and diverse 3D assets even from input prompts that contain complex compositional concepts (see Figure 1) and do not exist in the 3D dataset used for training. Due to its feed-forward architecture, Instant3D is exceptionally fast, requiring only about 20 seconds to generate a 3D asset, which is $200\times$ faster than previous optimization-based methods (Poole et al., 2022; Wang et al., 2023b) while achieving comparable or even better quality.

## 2 RELATED WORKS

**3D generation.** Following the success of generative models on 2D images using VAEs (Kingma & Welling, 2013; Van Den Oord et al., 2017), GANs (Goodfellow et al., 2014; Karras et al., 2019; Gu et al., 2022; Kang et al., 2023), and autoregressive models (Oord et al., 2016; Van Den Oord et al., 2016), people have also explored the applications of such models on 3D generation. Previous approaches have explored different methods to generate 3D models in the form of point clouds (Wu et al., 2016; Gadelha et al., 2017; Smith & Meger, 2017), triangle meshes (Gao et al., 2022; Pavllo et al., 2020; Chen et al., 2019; Luo et al., 2021) , volumes (Chan et al., 2022; Or-El et al., 2022; Bergman et al., 2022; Skorokhodov et al., 2022; Mittal et al., 2022) and implicit representations (Liu et al., 2022; Fu et al., 2022; Sanghi et al., 2022) in an unconditional or text/image-conditioned

manner. Such methods are usually trained on limited categories of 3D objects and do not generalize well to a wide range of novel classes.

Diffusion models (Rombach et al., 2021; Podell et al., 2023; Ho et al., 2020; Song et al., 2021; Saharia et al., 2022) open new possibilities for 3D generation. A class of methods directly train 3D diffusion models on the 3D representations (Nichol et al., 2022; Liu et al., 2023c; Zhou et al., 2021; Sanghi et al., 2023) or project the 3D models or multi-view rendered images into latent representations (Ntavelis et al., 2023; Zeng et al., 2022; Gupta et al., 2023; Jun & Nichol, 2023; Chen et al., 2023b) and perform the diffusion process in the latent space. For example, Shap-E (Jun & Nichol, 2023) encodes each 3D shape into a set of parameters of an implicit function, and then trains a conditional diffusion model on the parameters. These approaches face challenges due to the restricted availability and diversity of existing 3D data, consequently resulting in generated content with poor visual quality and inadequate alignment with the input prompt. Therefore, although trained on millions of 3D assets, Shap-E still fails to generate 3D shapes with complex compositional concepts and high-fidelity textures.

To resolve this, another line of works try to make use of 2D diffusion models to facilitate 3D generation. Some works (Jain et al., 2022; Mohammad Khalid et al., 2022) optimize meshes or NeRFs to maximize the CLIP Radford et al. (2021) score between the rendered images and input prompt utilizing pretrained CLIP models. While such methods can generate diverse 3D content, they exhibit a deficiency in visual realism. More recently, some works (Poole et al., 2022; Wang et al., 2023b; Lin et al., 2023; Chen et al., 2023c) optimize 3D representations using score distillation loss (SDS) based on pretrained 2D diffusion models. Such methods can generate high-quality results, but suffer from slow optimization, over-saturated colors and the Janus problem. For example, it takes 1.5 hours for DreamFusion (Poole et al., 2022) and 10 hours for ProlificDreamer Wang et al. (2023b) to generate a single 3D asset, which greatly limits their practicality. In contrast, our method enjoys the benefits of both worlds: it's able to borrow information from pretrained 2D diffusion models to generate diverse multi-view consistent images that are subsequently lifted to faithful 3D models, while still being fast and efficient due to its feed-forward nature.

**Sparse-view reconstruction.** Traditional 3D reconstruction with multi-view stereo (Agarwal et al., 2011; Schönberger et al., 2016; Furukawa et al., 2015) typically requires a dense set of input images that have significant overlaps to find correspondence across views and infer the geometry correctly. While NeRF (Mildenhall et al., 2020) and its variants (Müller et al., 2022; Chen et al., 2022; 2023a) have further alleviated the prerequisites for 3D reconstruction, they perform per-scene optimization that still necessitates a lot of input images. Previous methods (Wang et al., 2021; Chen et al., 2021; Long et al., 2022; Reizenstein et al., 2021; Trevithick & Yang, 2021; Shen et al., 2023) have tried to learn data priors so as to infer NeRF from a sparse set of images. Typically they extract per-view features from each input image, and then for each point on the camera ray, aggregate multi-view features and decode them to the density (or SDF) and colors. Such methods are either trained in a category-specific manner, or only trained on small datasets such as ShapeNet; they have not been demonstrated to generalize beyond these datasets especially to the complex text-to-2D outputs.

More recently, some methods utilize data priors from pretrained 2D diffusion models to lift a single 2D image to 3D by providing supervision at novel views using SDS loss (Liu et al., 2023b; Qian et al., 2023; Melas-Kyriazi et al., 2023) or generating multi-view images (Liu et al., 2023a). For instance, One-2-3-45 (Liu et al., 2023a) generates 32 images at novel views from a single input image using a fine-tuned 2D diffusion model, and reconstructs a 3D model from them, which suffers from inconsistency between the many generated views. In comparison, our sparse-view reconstructor adopts a highly scalable transformer-based architecture and is trained on large-scale 3D data. This gives it the ability to accurately reconstruct 3D models of novel unseen objects from a sparse set of 4 images without per-scene optimization.

## 3 METHOD

Our method Instant3D is composed of two stages: sparse-view generation and feed-forward NeRF reconstruction. In Section 3.1, we present our approach for generating sparse multi-view images conditioned on the text input. In Section 3.2, we describe our transformer-based sparse-view large reconstruction model.

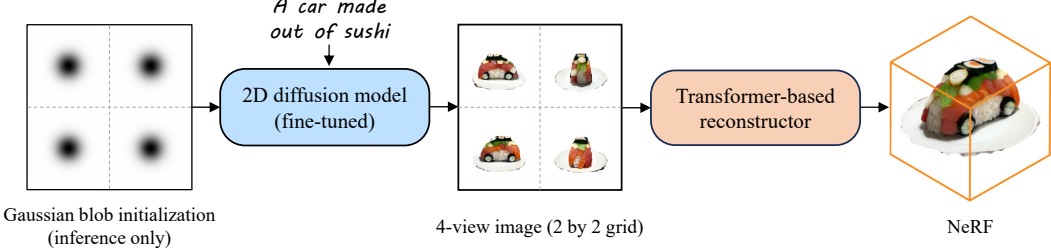

Figure 2: Overview of our method. Given a text prompt ('a car made out of sushi'), we perform multi-view generation with Gaussian blobs as initialization using fine-tuned 2D diffusion model, producing a 4-view image in the form of a $2 \times 2$ grid. Then we apply a transformer-based sparse-view 3D reconstructor on the 4-view image to generate the final NeRF.

## 3.1 TEXT-CONDITIONED SPARSE VIEW GENERATION

Given a text prompt, our goal is to generate a set of multi-view images that are aligned with the prompt and consistent with each other. We achieve this by fine-tuning a pretrained text-to-image diffusion model to generate a $2 \times 2$ image grid as shown in Figure 2.

In the following paragraphs, we first illustrate that large text-to-image diffusion models (i.e., SDXL (Podell et al., 2023)) have the capacity to generate view-consistent images thus a lightweight fine-tuning is possible. We then introduce three essential techniques to achieve it: the image grid, the curation of the dataset, and also the Gaussian Blob noise initialization in inference. As a result of these observations and technical improvements, we can fine-tune the 2D diffusion model for only 10K steps (on 10K data) to generate consistent sparse views.

**Multi-view generation with image grid.** Previous methods (Liu et al., 2023b;a) on novel-view synthesis show that image diffusion models are capable of understanding the multi-view consistency. In light of this, we compile the images at different views into a single image in the form of an image grid, as depicted in Figure 2. This image-grid design can better match the original data format of the 2D diffusion model, and is suitable for simple direct fine-tuning protocol of 2D models. We also observe that this simple protocol only works when the base 2D diffusion has enough capacity, as shown in the comparisons of Stable Diffusion v1.5 (Rombach et al., 2021) and SDXL (Podell et al., 2023) in Section 4.3. The benefit from simplicity will also be illustrated later in unlocking the lightweight fine-tuning possibility.

Regarding the number of views in the image grid, there is a trade-off between the requirements of multi-view generation and 3D reconstruction. More generated views make the problem of 3D reconstruction easier with more overlaps but increase possibility of view inconsistencies in generation and reduces the resolution of each generated view. On the other hand, too few views may cause insufficient coverage, requiring the reconstructor to hallucinate unseen parts, which is challenging for a deterministic 3D reconstruction model. Our transformer-based reconstructor learns generic 3D priors from large-scale data, and greatly reduces the requirement for the number of views. We empirically found that using 4 views achieves a good balance in satisfying the two requirements above, and they can be naturally arranged in a $2 \times 2$ grid as shown in Figure 2. Next, we detail how the image grid data is created and curated.

**Multi-view data creation and curation.** To fine-tune the text-to-image diffusion model, we create paired multi-view renderings and text prompts. We adopt a large-scale synthetic 3D dataset Objaverse (Deitke et al., 2023b) and render four $512 \times 512$ views of about $750K$ objects with Blender. We distribute the four views at a fixed elevation (20 degrees) and four equidistant azimuths (0, 90, 180, 270 degrees) to achieve a better coverage of the object. We use Cap3D (Luo et al., 2023) to generate captions for each 3D object, which consolidates captions from multi-view renderings generated with pretrained image captioning model BLIP-2 (Li et al., 2023) using a large language model (LLM). Finally, the four views are assembled into a grid image in a fixed order and resized to the input resolution compatible with the 2D diffusion model.

We find that naively using all the data for fine-tuning reduces the photo-realism of the generated images and thus the quality of the 3D assets. Therefore, we train a simple scorer on a small amount

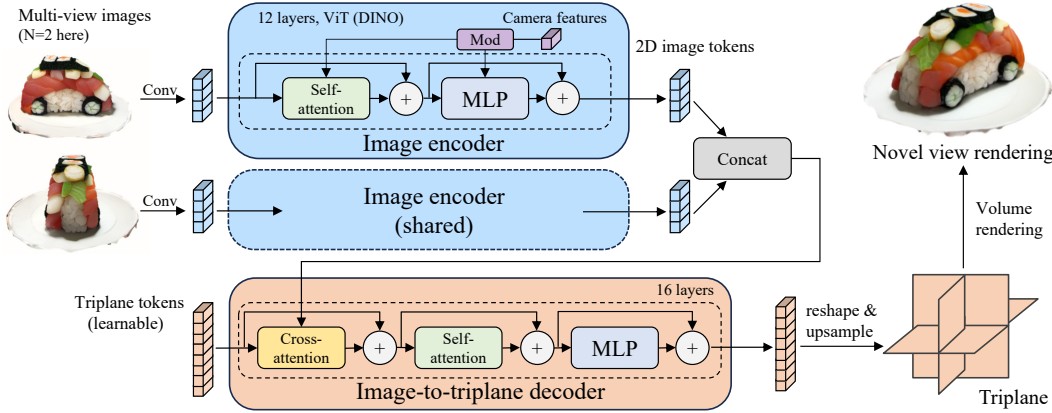

Figure 3: Architecture of our sparse-view reconstructor. The model applies a pretrained ViT to encode multi-view images into pose-aware image tokens, from which we decode a triplane representation of the scene using a transformer-based decoder. Finally we decode per-point triplane features to its density and color and perform volume rendering to render novel views. We illustrate here with 2 views and the actual implementation uses 4 views.

(2000 samples) of manually labeled data to predict the quality of each 3D object. The model is a simple SVM on top of pretrained CLIP features extracted from multi-view renderings of the 3D object (please see Appendix for details). During training, our model only takes the top 10K data ranked by our scorer. We provide a quantitative study in Section 4.3 to validate the impact of different data curation strategies. Although the difference is not very significant from the metric perspective, we found that our curated data is helpful in improving the visual quality.

**Inference with Gaussian blob initialization.** While our training data is multi-view images with a white background, we observe that during inference starting from standard Gaussian noise still results in images that have cluttered backgrounds (see Figure 5); this introduces extra difficulty for the feed-forward reconstructor in the second stage (Section 3.2). To guide the model toward generating images with a clean white background, inspired by SDEdit (Meng et al., 2022), we first create an image of a $2 \times 2$ grid with a solid white background that has the same resolution as the output image, and initialize each sub-grid with a 2D *Gaussian blob* that is placed at the center of the image with a standard deviation of 0.1 (please see Appendix for details). The visualization of this Gaussian Blob is shown in Figure 2. The Gaussian blob image grid is fed to the auto-encoder to get its latent. We then add diffusion noise (e.g., use t=980/1000 for 50 DDIM denoising steps), and use it as the starting point for the denoising process. As seen in Figure 5, this technique effectively guides the model toward generating images with a clean background.

**Lightweight fine-tuning.** With all the above observations and techniques, we are able to adapt a text-to-image diffusion model to a text-to-multiview model with lightweight fine-tuning. This lightweight fine-tuning shares a similar spirit to the 'instruction fine-tuning' (Mishra et al., 2022; Wei et al., 2021) for LLM alignment. The assumption is that the base model is already capable of the task, and the fine-tuning is to unlock the base model's ability without introducing additional knowledge.

Since we utilize an image grid, the fine-tuning follows the exactly same protocol as the 2D diffusion model pre-training, except that we decrease the learning rate to $10^{-5}$. We train the model with a batch size of 192 for only 10K iterations on the 10K curated multi-view data. The training is done using 32 NVIDIA A100 GPUs for only 3 hours. We study the impact of different training settings in Section 4.3. For more training details, please refer to Appendix.

## 3.2 FEED-FORWARD SPARSE-VIEW LARGE RECONSTRUCTION MODEL

In this stage, we aim to reconstruct a NeRF from the four-view images $\mathcal{I} = \{\mathbf{I}_i \mid i = 1, ..., 4\}$ generated in the first stage. 3D reconstruction from sparse inputs with a large baseline is a challeng-

ing problem, which requires strong model priors to resolve the inherent ambiguity. Inspired by a recent work LRM (Hong et al., 2024) that introduces a transformer-based model for single image 3D reconstruction, we propose a novel approach that enables us to predict a NeRF from a sparse set of input views with known poses. Similar to Hong et al. (2024), our model consists of an image encoder, an image-to-triplane decoder, and a NeRF decoder. The image encoder encodes the multi-view images into a set of tokens. We feed the concatenated image tokens to the image-to-triplane decoder to output a triplane representation (Chan et al., 2022) for the 3D object. Finally, the triplane features are decoded into per-point density and colors via the NeRF MLP decoder.

In detail, we apply a pretrained Vision Transformer (ViT) DINO (Caron et al., 2021) as our image encoder. To support multi-view inputs, we inject camera information in the image encoder to make the output image tokens pose-aware. This is different from Hong et al. (2024) that feeds the camera information in the image-to-triplane decoder because they take single image input. The camera information injection is done by the AdaLN (Huang & Belongie, 2017; Peebles & Xie, 2022) camera modulation as described in Hong et al. (2024). The final output of the image encoder is a set of pose-aware image tokens $\boldsymbol{f}_{\mathbf{I}_i}^*$, and we concatenate the per-view tokens together as the feature descriptors for the multi-view images: $\boldsymbol{f}_{\mathcal{I}} = \oplus(\boldsymbol{f}_{\mathbf{I}_1}^*, ... \boldsymbol{f}_{\mathbf{I}_4}^*)$

We use triplane as the scene representation. The triplane is flattened to a sequence of learnable tokens, and the image-to-triplane decoder connects these triplane tokens with the pose-aware image tokens $\boldsymbol{f}_{\mathcal{I}}$ using cross-attention layers, followed by self-attention and MLP layers. The final output tokens are reshaped and upsampled using a de-convolution layer to the final triplane representation. During training, we ray march through the object bounding box and decode the triplane features at each point to its density and color using a shared MLP, and finally get the pixel color via volume rendering. We train the networks in an end-to-end manner with image reconstruction loss at novel views using a combination of MSE loss and LPIPS (Zhang et al., 2018) loss.

**Training details.** We train the model on multi-view renderings of the Objaverse dataset (Deitke et al., 2023b). Different from the first stage that performs data curation, we use all the 3D objects in the dataset and scale them to $[-1, 1]^3$; then we generate multi-view renderings using Blender under uniform lighting with a resolution of $512 \times 512$. While the output images from the first stage are generated in a structured setup with fixed camera poses, we train the model using random views as a data augmentation mechanism to increase the robustness. Particularly, we randomly sample 32 views around each object. During training, we randomly select a subset of 4 images as input and another random set of 4 images as supervision. For inference, we will reuse the fixed camera poses in the first stage as the camera input to the reconstructor. For more details on the training, please refer to the Appendix.

## 4 EXPERIMENTS

In this section, we first do comparisons against previous methods on text-to-3D (Section 4.1), and then perform ablation studies on different design choices of our method. By default, we report the results generated with fine-tuned SDXL models, unless otherwise noted.

### 4.1 TEXT-TO-3D

We make comparisons to state-of-the-art methods on text-to-3D, including a feed-forward method Shap-E (Jun & Nichol, 2023), and optimization-based methods including DreamFusion (Poole et al., 2022) and ProlificDreamer (Wang et al., 2023b). We use the official code for Shap-E, and the implementation from three-studio (Guo et al., 2023) for the other two as there is no official code. We use default hyper-parameters (number of optimization iterations, number of denoising steps) of these models. For our own model we use the SDXL base model fine-tuned on 10K data for 10K steps. During inference we take 100 DDIM steps.

**Qualitative comparisons.** As shown in Figure 4, our method generates visually better results than those of Shap-E, producing sharper textures, better geometry and substantially improved text-3D alignment. Shap-E applies a diffusion model that is exclusively trained on million-level 3D data, which might be evidence for the need of 2D data or models with 2D priors. DreamFusion and ProlificDreamer achieve better text-3D alignment utilizing pretrained 2D diffusion models. However,

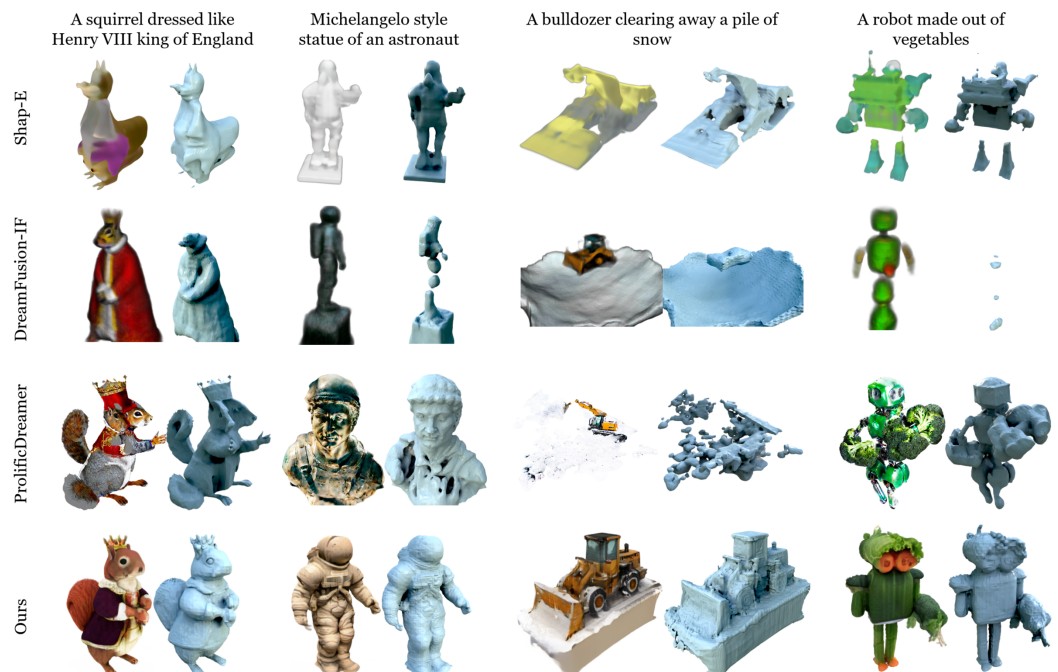

Figure 4: Qualitative comparisons on text-to-3D against previous methods. We include more uncurated comparisons in the supplementary material.

Table 1: Quantitative comparisons on CLIP scores against baseline methods. Our method outperforms previous feed-forward method Shap-E and optimization-based method DreamFusion, and achieves competitive performance compared to ProlificDreamer while being $1800\times$ faster.

Table 2: Quantitative comparisons against previous sparse-view reconstruction methods on GSO dataset.

|  | ViT-L/14 ↑ | ViT-bigG-14 ↑ | Time(s) ↓ |
|---|---|---|---|
| Shap-E | 20.51 | 32.21 | 6 |
| DreamFusion | 23.60 | 37.46 | 5400 |
| ProlificDreamer | 27.39 | 42.98 | 36000 |
| Ours | 26.87 | 41.77 | 20 |

|  | PSNR ↑ | SSIM ↑ | LPIPS ↓ |
|---|---|---|---|
| SparseNeus | 20.62 | 0.8360 | 0.1989 |
| Ours | 26.54 | 0.8934 | 0.0643 |

DreamFusion generates results with over-saturated colors and over-smooth textures. While ProlificDreamer results have better details, it still suffers from low-quality geometry (as in 'A bulldozer clearing ...') and the Janus problem (as in "a squirrel dressed like ...", also more detailed in Appendix Figure 11). In comparison, our results have more photorealistic appearance with better geometric details. Please refer to the Appendix and supplementary materials for video comparisons and more results.

**Quantitative comparisons.** In Table 4, we quantitatively assess the coherence between the generated models and text prompts using CLIP-based scores. We perform the evaluation on results with 400 text prompts from DreamFusion. For each model, we render 10 random views and calculate the average CLIP score between the rendered images and the input text. We report the metric using multiple variants of CLIP models with different model sizes and training data (i.e., ViT-L/14 from OpenAI and ViT-bigG-14 from OpenCLIP). From the results we can see that our model achieves higher CLIP scores than Shap-E, indicating better text-3D alignment. Our method even achieves consistently higher CLIP scores than optimization-based method DreamFusion and competitive scores to ProlificDreamer, from which we can see that our approach can effectively inherit the great text understanding capability from the pretrained SDXL model and preserve them in the generated 3D assets via consistent sparse-view generation and robust 3D reconstruction.

**Inference time comparisons.** We present the time to generate a 3D asset in Table 1. The timing is measured using the default hyper-parameters of each method on an A100 GPU. Notably, our method is significantly faster than the optimization-based methods: while it takes 1.5 hours for DreamFusion and 10 hours for ProlificDreamer to generate a single asset, our method can finish the generation within 20 seconds, resulting in a $270\times$ and $1800\times$ speed up respectively. In Figure 10, we show that our inference time can be further reduced without obviously sacrificing the quality by decreasing the number of DDIM steps.

## 4.2 COMPARISONS ON SPARSE VIEW RECONSTRUCTION

We make comparisons to previous sparse-view NeRF reconstruction works. Most of previous works (Reizenstein et al., 2021; Trevithick & Yang, 2021; Yu et al., 2021) are either trained on small-scale datasets such as ShapeNet, or trained in a category-specific manner. Therefore, we make comparisons to a state-of-the-art method SparseNeus (Long et al., 2022), which is also applied in One-2-3-45 (Liu et al., 2023a) where they train the model on the same Objaverse dataset for sparse-view reconstruction. We do the comparisons on the Google Scan Object (GSO) dataset (Downs et al., 2022), which consists of 1019 objects. For each object, we render 4-view input following the structured setup and randomly select another 10 views for testing. We adopt the pretrained model from Liu et al. (2023a). Particularly, SparseNeus does not work well for 4-view inputs with such a large baseline; therefore we add another set of 4 input views in addition to our four input views (our method still uses 4 views as input), following the setup in Liu et al. (2023a). We report the metrics on novel view renderings in Table 2. From the table, we can see that our method outperforms the baseline method even with fewer input images, which demonstrates the superiority of our sparse-view reconstructor.

## 4.3 ABLATION STUDY FOR SPARSE VIEW GENERATION

We ablate several key decisions in our method design, including (1) the choice of the larger 2D base model SDXL, (2) the use of Gaussian Blob during inference, (3) the quality and size of the curated dataset, and lastly, (4) the need and requirements of lightweight fine-tuning. We gather the quantitative results in Table 3 and place all qualitative results in the Appendix. We observe that qualitative results are more evident than quantitative results, thus we recommend a closer examination.

**Scalability with 2D text-to-image models.** One of the notable advantages of our method is that its efficacy scales positively with the potency of the underlying 2D text-to-image model. In Figure 12, we present qualitative comparisons between two distinct backbones (with their own tuned hyper-parameters): SD1.5 (Rombach et al., 2021) and SDXL (Podell et al., 2023). It becomes readily apparent that SDXL, which boasts a model size $3\times$ larger than that of SD1.5, exhibits superior text comprehension and visual quality. We also show a quantitative comparison on CLIP scores in Table 3. By comparing Exp(l, m) with Exp(d, g), we can see that the model with SD1.5 achieves consistently lower CLIP scores indicating worse text-3D alignment.

**Gaussian blob initialization.** In Figure 5, we show our results generated with and without Gaussian blob initialization. From the results we can see that while our fine-tuned model can generate multi-view images without Gaussian blob initialization, they tend to have cluttered backgrounds, which challenges the second-stage feed-forward reconstructor. In contrast, our proposed Gaussian blob initialization enables the fine-tuned model to generate images with a clean white background, which better align with the requirements of the second stage.

**Quality and size of fine-tuning dataset.** We evaluate the impact of the quality and size of the dataset used for fine-tuning 2D text-to-image models. We first make comparisons between curated and uncurated (randomly selected) data. The CLIP score rises slightly as shown in Table 3 (i.e., comparing Exp(d, i)), while there is a substantial quality improvement as illustrated in Appendix Figure 7. This aligns with the observation that the data quality can dramatically impact the results in the instruction fine-tuning stage of LLM (Zhou et al., 2023).

When it comes to data size, we observe a double descent from Table 3 Exp(a, d, g) with 1K, 10K, and 100K data. We pick Exp(a, d, g) here because they are the best results among different training steps for the same training data size. The reason for this double descent can be spotlighted by the

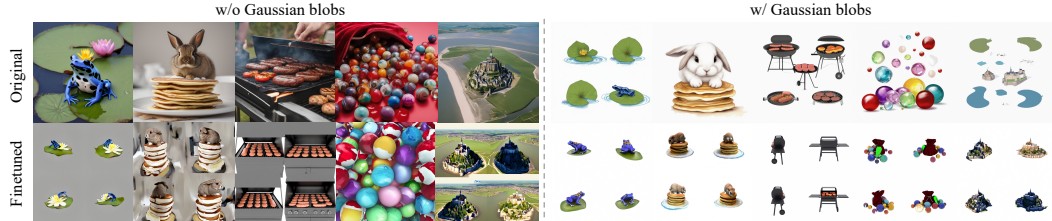

Figure 5: Qualitative comparisons on results generated with and without Gaussian blob initialization.

Table 3: Comparison on CLIP scores of NeRF renderings with different variants of fine-tuning settings.

| Exp ID | Exp Name | Base | # Data | Curated | # Steps | ViT-L/14 | ViT-bigG-14 |
|--------|----------|------|--------|---------|---------|----------|-------------|
| (a) | Curated-1K-s1k | SDXL | 1K | ✓ | 1K | 26.33 | 41.09 |
| (b) | Curated-1K-s10k | SDXL | 1K | ✓ | 10k | 22.55 | 35.59 |
| (c) | Curated-10K-s4k | SDXL | 10K | ✓ | 4k | 26.55 | 41.08 |
| (d) | Curated-10K-s10k | SDXL | 10K | ✓ | 10k | **26.87** | **41.77** |
| (e) | Curated-10K-s20k | SDXL | 10K | ✓ | 20k | 25.96 | 40.56 |
| (f) | Curated-100K-s10k | SDXL | 100K | ✓ | 10k | 25.79 | 40.32 |
| (g) | Curated-100K-s40k | SDXL | 100K | ✓ | 40k | 26.59 | 41.29 |
| (h) | Curated-300K-s40k | SDXL | 300K | ✓ | 40K | 26.43 | 40.72 |
| (i) | Random-10K-s10k | SDXL | 10K | ✗ | 10k | 26.87 | 41.47 |
| (j) | Random-100K-s40k | SDXL | 100K | ✗ | 40k | 26.28 | 40.90 |
| (k) | AllData-s40k | SDXL | 700K | ✗ | 40k | 26.13 | 40.60 |
| (l) | Curated-10K-s10k (SD1.5) | SD1.5 | 10K | ✓ | 10k | 23.50 | 36.90 |
| (m) | Curated-100K-s40k (SD1.5) | SD1.5 | 100K | ✓ | 40k | 25.48 | 39.07 |

qualitative comparisons in Appendix Figure 13, where training with 1K data can lead to inconsistent multi-view images, while training with 100K data can hurt the compositionality, photo-realism, and also text alignment.

**Number of fine-tuning steps.**  We also quantitatively and qualitatively analyze the impact of fine-tuning steps. For each block in Table 3 we show the CLIP scores of different training steps. Similar to the findings in instruction fine-tuning (Ouyang et al., 2022), the results do not increase monotonically regarding the number of fine-tuning steps but have a peak in the middle. For example, in our final setup with the SDXL base model and 10K curated data (i.e., Exp(c, d, e)), the results are peaked at 10K steps. For other setups, the observations are similar. We also qualitatively compare the results at different training steps for 10K curated data in Appendix Figure 14. There is an obvious degradation in the quality of the results for both 4K and 20K training steps.

Another important observation is that the peak might move earlier when the model size becomes larger. This can be observed by comparing between Exp(l,m) for SD1.5 and Exp(d,g) for SDXL. Note that this comparison is not conclusive yet from the Table given that SD1.5 does not perform reasonably with our direct fine-tuning protocol. More details are in the Appendix.

We also found that Exp(a) with 1K steps on 1K data can achieve the best CLIP scores but the view consistency is actually disrupted. A possible reason is that the CLIP score is insensitive to certain artifacts introduced by reconstruction from inconsistent images, which also calls for a more reliable evaluation metric for 3D generation.

## 5  CONCLUSIONS

In this paper we presented a novel feed-forward two-stage approach Instant3D that can generate high-quality and diverse 3D assets from text prompts within 20 seconds. Our method finetunes a 2D text-to-image diffusion model to generate consistent 4-view images, and lifts them to 3D with a robust transformer-based large reconstruction model. The experiment results show that our method outperforms previous feed-forward methods in terms of quality while being equally fast, and achieves comparable or better performance to previous optimization-based methods with a speed-up of more than 200 times. Instant3D allows novice users to easily create 3D assets and enables fast prototyping and iteration for various applications such as 3D design and modeling.

**Ethics Statement.** The generation ability of our model is inherited from the public 2D diffusion model SDXL. We only do lightweight fine-tuning over the SDXL model thus it is hard to introduce extra knowledge to it. Also, our model can share similar ethical and legal considerations to SDXL. The curation of the data for lightweight fine-tuning does not introduce outside annotators. Thus the quality of the data might be biased towards the preference of the authors, which can lead to a potential bias on the generated results as well. The text input to the model is not further checked by the model, which means that the model will try to do the generation for every text prompt it gets without the ability to acknowledge unknown knowledge.

**Reproducibility Statement.** In the main text, we highlight the essential techniques to build our model for both the first stage (Section 3.1) and the second stage (Section 3.2). We discuss how our data is created and curated in Section 3. The full model configurations and training details can be found in Appendix Section A.3 and Section A.6. We have detailed all the optimizer hyper-parameters and model dimensions. We present more details on our data curation process in Section A.2. We also attach the IDs of our curated data in Supplementary Materials to further facilitate the reproduction.

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

# A APPENDIX

## A.1 DIVERSITY OF GENERATION

Inheriting the generation capability from the base SDXL model, our method can generate diverse results from the same text prompt by using different random seeds in the feed-forward pass. As shown in Figure 6, our approach excels in generating diverse 3D assets featuring strikingly distinct textures and geometries from the same prompt. This is in contrast to previous SDS-optimization based methods, which are prone to generate similar results even with different initializations (Poole et al., 2022).

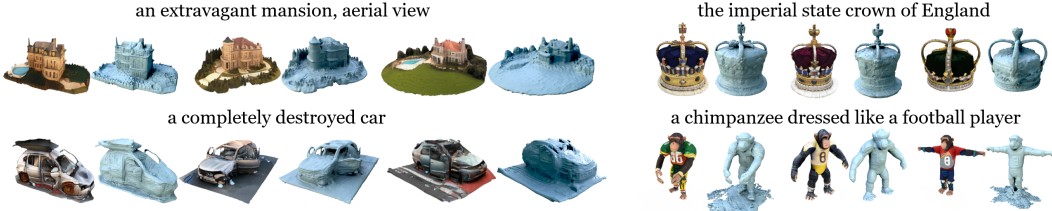

Figure 6: Our method can generate diverse results from the same text prompt.

## A.2 DATA CURATION DETIALS

We apply a quality scorer to curate high-quality data from the Objaverse dataset. To train the quality scorer, we first randomly sample 2000 3D objects from the dataset and manually label each 3D asset as good or bad. Good assets have realistic textures and complex geometry, while bad ones have simple shapes and flat or cartoon-like textures. This criterion is subjective and imprecise, but we found it good enough for the purpose of data filtering.

Since the amount of annotated data is limited, we use a pretrained CLIP (Radford et al., 2021) model to extract high-level image features of rendered images at 5 randomly sampled camera viewpoints for each object. Then we train a simple binary SVM classifier on top of the averaged CLIP features over different views. We use the NuSVC implementation from the popular scikit-learn framework Pedregosa et al. (2011), which also gives us a probability estimation of the classification. We use the trained SVM model to predict the classification probability for all objects in the dataset by extracting CLIP features in the same way as done for the training data. These probabilities are used as scores to rank the data from high to low quality. Finally, we use the top 10K objects as our fine-tuning data.

To render the 4-view data, we scale the curated objects and center them at a cube $[-1, 1]^3$. We render the objects with a white background following the structured setup discussed in Section 3.1 using a field of view $50°$ at a distance of 2.7 under uniform lighting. We use the physically-based path tracer Cycles in Blender for rendering.

In Figure 7 we show qualitative comparisons on results from models trained with curated data and random data. Models trained with random data tend to generate cartoon-like 3D assets with simple and flat textures. This is not surprising since a bulk of the Objaverse dataset contains simple shapes with simple textures, and without curation these data will guide the model to over-denoise the results, leading to large areas of flat colors. On the contrary, models trained with curated data tend to generate more photorealistic assets with complex textures and geometries.

## A.3 SDXL FINE-TUNING DETAILS

We use SDXL as the base model for our first-stage fine-tuning. We use AdamW optimizer with a fixed learning rate $10^{-5}$, $\beta_1 = 0.9$, $\beta_2 = 0.999$ and a weight decay of $10^{-2}$. We fine-tune the model using fp16 on 32 NVIDIA A100 GPUs with a total batch size of 192. No gradient accumulation is used. We train the model on 10K curated data for 40K steps, which takes around 3 hours.

We train the model with the standard denoising diffusion loss (Ho et al., 2020)

$$L(\boldsymbol{\theta}) = \mathbb{E}_{t,\boldsymbol{x}_0,\boldsymbol{\epsilon}}\big[\|\boldsymbol{\epsilon} - \boldsymbol{\epsilon_\theta}(\sqrt{\overline{\alpha}_t}\boldsymbol{x}_0 + \sqrt{1-\overline{\alpha}_t}\boldsymbol{\epsilon}, t)\|^2\big] \tag{1}$$

where $\boldsymbol{\epsilon_\theta}$ is the denoising U-Net and $\boldsymbol{\theta}$ are the trainable parameters.

SDXL introduces image resolution and aspect ratio conditioning that allow mixing training on images of different resolutions and aspect ratios. As for our training data, we render 4 views each with a resolution of $512 \times 512$ and assemble them into a $1024 \times 1024$ image. Therefore we fix the resolution and aspect ratio conditioning to be $(1024, 1024)$ throughout the fine-tuning procedure. We don't do random cropping in our training and fixed the crop conditioning to be $(0, 0)$. All the other training setups are identical to the original SDXL.

Table 4: Ablation study of the sparse-view reconstruction model.

| | #Layers | Render | Supervision | PSNR ↑ | SSIM ↑ | LPIPS ↓ |
|---|---|---|---|---|---|---|
| exp01 | 6 | 64 | All | 23.6551 | 0.8616 | 0.1281 |
| exp02 | 12 | 64 | All | 23.8257 | 0.8631 | 0.1266 |
| exp03 | 24 | 64 | All | 23.8351 | 0.8635 | **0.1258** |
| exp04 | 12 | 32 | All | 23.1704 | 0.8561 | 0.1358 |
| exp05 | 12 | 64 | w/o novel | 18.2359 | 0.8103 | 0.2256 |
| exp06 | 12 | 64 | w/o LPIPS | **24.1699** | **0.8641** | 0.1934 |

## A.4 SD1.5 FINE-TUNING DETAILS

We use 8 A100 GPUs for fine-tuning SD1.5 on 100K data with a total batch size of 64. We use the same AdamW optimizer as the one for SDXL with the same hyper-parameters. We also use gradient accumulation of 3 steps, which gives an effective batch size of 192. The training loss is the same as SDXL. We train the model for 120K steps (40K parameter updates due to gradient accumulation), which takes roughly 33 hours.

## A.5 GAUSSIAN BLOBS INITIALIZATION

Since the diffusion model is fine-tuned with only a relatively small number of steps, it still largely possesses the original denoising behavior on images that are not in the form of $2 \times 2$ grids and do not have a white background. Naively applying the standard backward denoising process starting from random Gaussian noise will likely lead to results far from the distribution of the fine-tuning data (see Figure 5).

The spatial structure of the training images is simple: four views of the same object are placed at the center of each quadrant. Also, the background is always white. Since the model is fine-tuned on such data with a denoising objective, it is natural that, when presented with a noisy input whose underlying clean image has these two characteristics, the model will tend to denoise the image to a clean one where the four-quadrant objects are view consistent. Following this, and inspired by SDEdit Meng et al. (2022), we introduce Gaussian blobs initialization to guide the model toward generating samples consistent with the distribution of the fine-tuning data.

The standard latent diffusion inference starts with a Gaussian noise image $\epsilon$ with the same size as the image latents. Instead, we modify the initial iteration to be a composition of Gaussian noise and an image with the two aforementioned characteristics: object quadrants and white background. We construct such an image by generating a grayscale image with a clean white background and a black Gaussian blob at the center. Specifically, we construct a $H \times W$ grayscale image $I$, where $H$ and $W$ are the height and width of the input RGB image with a value range $[0, 1]$. For all our models $H = W$, and we denote them using $S$. For a given pixel $(x, y)$, its pixel value is computed as

$$I(x, y) = 1 - \exp\left(-\frac{(x - S/2)^2 + (y - S/2)^2}{2\sigma^2 S^2}\right) \qquad (2)$$

where $\sigma$ is a hyper-parameter controlling the width of the Gaussian blob. Such an image looks like a black disc at the center of a white image slowly fading away toward the edges of the image. We then assemble four such images into a $2 \times 2$ image grid. Some examples of such images with different $\sigma$ can be seen at the first row of figure 5.

Next we construct the initial noise for the denoising step by blending a complete Gaussian noise latent with the latent of the Gaussian blobs. We denote the latent of the Gaussian blobs image $I$ as $\tilde{I}$, and the latent of a noise image with i.i.d. Gaussian values as $\epsilon$. For a $N$ step denoising inference process with timesteps $\{t_N, t_{N-1}, ..., t_0\}$, we mix the two latents with a weighted sum

$$\epsilon_{t_N} = \sqrt{\overline{\alpha}_{t_N}} \tilde{I} + \sqrt{1 - \overline{\alpha}_{t_N}} \epsilon \qquad (3)$$

Then $\epsilon_{t_N}$ is used as the initial noise of the denoising process, e.g., $t_N$ is 980 for a denoising step with 50 (and the total number of timesteps is 1000).

### A.6 SPARSE-VIEW RECONSTRUCTION DETAILS

**Model details**  We use the DINO-ViT-B/16 as our image encoder. This model is transformer-based, which has 12 layers and the hidden dimension of the transformer is 768. The ViT begins with a convolution of kernel size 16, stride 16, and padding 0. It is essentially patchifying the input image with a patch size of $16 \times 16$. For our final model, the input image resolution is 512, thus it leads to $32 \times 32 = 1024$ spatial tokens in the vision transformer. In ablation studies, we reduce the input resolution from 512 to 256 to save compute budget. The original DINO is trained with a resolution of $224 \times 224$, thus the positional embedding has only a size of $14 \times 14 = 196$. We thus use 2D bilinear extrapolation (with `torch.nn.functional.interpolate` function) to extrapolate it to the desired token size.

To integrate camera information into the image encoder, we inject modulation layers (Peebles & Xie, 2022) into each of the transformer layer (for both self-attention layers and MLP layers). The modulation layer is initialized to be an identity mapping and thus it is suitable to be added to a pre-trained vision transformer.

After the image encoder, we have 1025 image feature tokens for each image, since we also include the output of the `[CLS]` token. We concatenate the tokens from all four images to construct a sequence of condition features of length 4100. This condition feature will be used to create the keys and values in the cross-attention layers of the image-to-triplane transformer decoder.

The image-to-triplane transformer decoder starts with a token sequence of $(3 \times 32 \times 32) \times 1024$, where $(3 \times 32 \times 32)$ is the number of tokens and $1024$ is the hidden dimension of the transformer. We use 16 layers in our transformer decoder. All attention layers have 16 attention heads and each head has a dimension of 64. We remove the bias term in the attention layer as in Touvron et al. (2023). We take the pre-normalization architecture of the transformer where each sub-layer will be in the format of $x + f(\text{LayerNorm}(x))$.

After the transformer, we apply a de-convolution layer to map the transformer output from $(3 \times 32 \times 32) \times 1024$ to $3 \times (64 \times 64) \times 80$. It means that there are 3 planes (XY, YZ, XZ) (Chan et al., 2022) and each plane has a size of $64 \times 64$. The dimension of each plane is $80$. All three planes share the same deconvolution layer. The deconvolution is of kernal size 2, stride 2, and pad 0.

In NeRF volumetric rendering, the features from the three planes are bilinearly interpolated and concatenated to get a 240-dimensional feature for each point. Then, we have a 10-layer MLP with a hidden dimension of 64 to map this 240-dim feature to a 4-dim feature. The first three dimensions will be treated as RGB colors of the point and normalized to [0, 1] with a sigmoid function. The last dimension will be treated as the density value and we use an exponential function to map the MLP's output to be non-negative.

For the exact formulation of the above operators, please refer to LRM (Hong et al., 2024) and DiT (Peebles & Xie, 2022).

**Training details.**  We adopt the AdamW (Kingma & Ba, 2014; Loshchilov & Hutter, 2017) optimizer to train our model. We use a peak learning rate of $4 \times 10^{-4}$ with a linear warm-up (on the first 3K steps) and a cosine decay. We change the $\beta_2$ of the AdamW optimizer to $0.95$ for better stability. We use a weight-decay of 0.05 for non-bias and non-layernorm parameters. We also apply a gradient clipping of 1.

For the initialization of the image encoder, we use the official DINO pre-trained weight. For the initialization of the triplane decoder, and NeRF MLP, we use the default initializer in the PyTorch implementation. We empirically found that the pre-normalization transformer is robust to different initialization of linear layers. For the positional embedding of the triplane tokens in the transformer decoder, we initialize them with a Gaussian of zero-mean and std of $1/\sqrt{1024}$.

For each training step, we randomly sample 4 views as input and another 4 as supervision. The number of sample points per ray in NeRF rendering is 128, which are uniformly distributed along the segment within the $[-1, 1]^3$ bounding box. The rendering resolution is $128 \times 128$. To allow higher actual supervising resolution, we first resize the image to a smaller resolution (uniformly sampled from [128, 384]) and then crop a patch of $128 \times 128$ from it. Thus we can go beyond the rendering resolution of 128.

We utilize flash attention (Dao et al., 2022), mixed-precision training (with bf16 as the half-precision format) (Micikevicius et al., 2018), and gradient checkpointing (Chen et al., 2016) to improve the compute/memory efficiency of the training.

We perform the training for 120 epochs on our rendered Objaverse data with a training batch size of 1024. We use both L2 loss and LPIPS loss to supervise the model and the weights of the two losses are 1 and 2 respectively. The model is trained on 128 NVIDIA A100 GPUs and the whole training can be finished in 7 days.

## A.7   SPARSE VIEW RECONSTRUCTION ABLATION STUDY

We conduct an ablation study of our sparse-view reconstruction model to validate different design choices including the number of layers in the image-to-triplane decoder, the rendering resolution and the losses used during training, and the usage of novel view supervision. We train the model on the same dataset as our final model, however, we change the training recipe to reduce the computation cost to 32 A100 GPUs for 1 day. The changes of configuration for ablation include (1) a resolution of $256 \times 256$ for the input image resolution, (2) 96 points per ray during rendering, (3) 5 layers instead of 10 layers in the NeRF MLP, (4) 30 epochs of training.

To evaluate the performance of different variants, we test them on another 3D dataset Google Scanned Object (GSO) (Downs et al., 2022). For each object in GSO, we render a set of 64-view images rendered with a resolution of $512 \times 512$ at elevations $0°$, $20°$, $40°$, $60°$. Each elevation has 16 views with equidistant azimuths starting from 0. We use 4 views with elevation $20°$ and azimuths $45°$, $135°$, $225°$, $315°$ as input, and randomly sample 5 views from the remaining views as our testing set, which stay the same for different variants. We render the 5 testing views and report their difference from the ground truth using 3 metrics including PSNR, SSIM and LPIPS. These metrics are averaged over all 1019 objects in the GSO dataset.

The results of the ablation studies are in Figure 4. From the table we can see that the model is robust to the number of transformer layers in the image-to-triplane decoder as shown in exp01, exp02, and exp03. We also observe that the LPIPS loss can largely affect the results by comparing the exp02 and exp06. Without the LPIPS loss, the model drops a lot on the LPIPS metric while getting a slight improvement on PSNR and SSIM. However, we empirically find that LPIPS is much more aligned with human perception and the rendered images become blurry without it. The rendering resolution is also important (as shown in exp04) since LPIPS can be more robust and accurate at a higher resolution, which also motivates us to use a rendering resolution of 128 by 128 when training our final model.

Also, the inclusion of novel view supervision in the training is critical as shown in exp05. All three metrics got a significant drop when only supervising the four input views. Upon reviewing the results, we find that it's due to the insufficient coverage of the four views, which typically leads to floaters in regions not covered by the input views.

## A.8   EXTENSION TO IMAGE-CONDITIONED GENERATION

Our method can also be extended to support additional image conditioning to provide more fine-grained control over the 3D model to be generated. In this process, the input to the model includes an input text prompt that describes the object to be generated as well as an image of the object. We use the same training data as our text-conditioned model. During training, for a randomly sampled time step, we keep the latent of the input image (top-left quadrant) untouched and only add noise to the latents of the remaining three views. This allows the diffusion model to generate the other views while accounting for the conditioning image. During inference, similarly, we replace the upper left quadrant of the latent feature with the latent of the clean conditioning image at each iteration. Figure 8 shows some visual results of our image-conditioned model. From the results we can see that our method is able to effectively generate the other views with faithful details that are coherent with the input text prompt and image, thus giving us high-quality 3D models with our sparse view reconstructor.

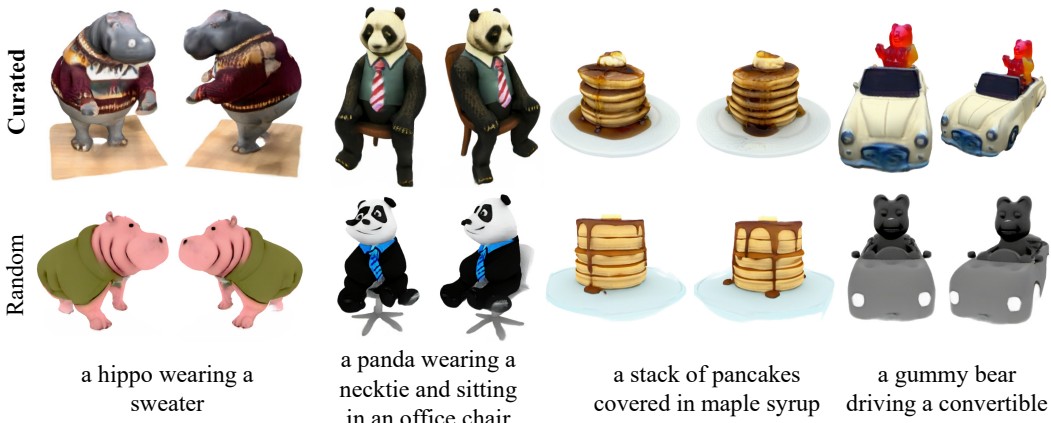

**Curated**

**Random**

a hippo wearing a sweater

a panda wearing a necktie and sitting in an office chair

a stack of pancakes covered in maple syrup

a gummy bear driving a convertible

Figure 7: Comparisons on novel view renderings of NeRF assets generated from SDXL models fine-tuned with 10K curated data and random data. We can see that that curated data enables the model to generate more photorealistic 3D assets with more geometric and texture details. Here curated and random correspond to Exp d (Curated-10K-s10K) and i (Random-10K-s10K) in Table 3.

## A.9 LIMITATIONS

While our model can generate high-quality and diverse 3D assets, it still suffers from several limitations. First, while we perform a light-weight fine-tuning that enables the model to mostly preserve the capability of the SDXL model in textual understanding and generation, we do observe that our model fails to handle some over-complicated prompts, for example, those related to complex spatial arrangements of multiple subjects and complex scenes (see Figure 15). In addition, the generated assets are not as photorealistic as the 2D images generated by the original SDXL, which may be attributed to the information loss in the fine-tuning stage. Secondly, there is a lack of 3D inductive bias when generating multi-view images, and therefore it's still possible for our model to generate inconsistent images that result in low-quality 3D assets with corrupted geometries and textures. Finally, our feed-forward reconstructor tends to generate blurry textures compared to the input images due to the usage of a relatively low-resolution triplane.

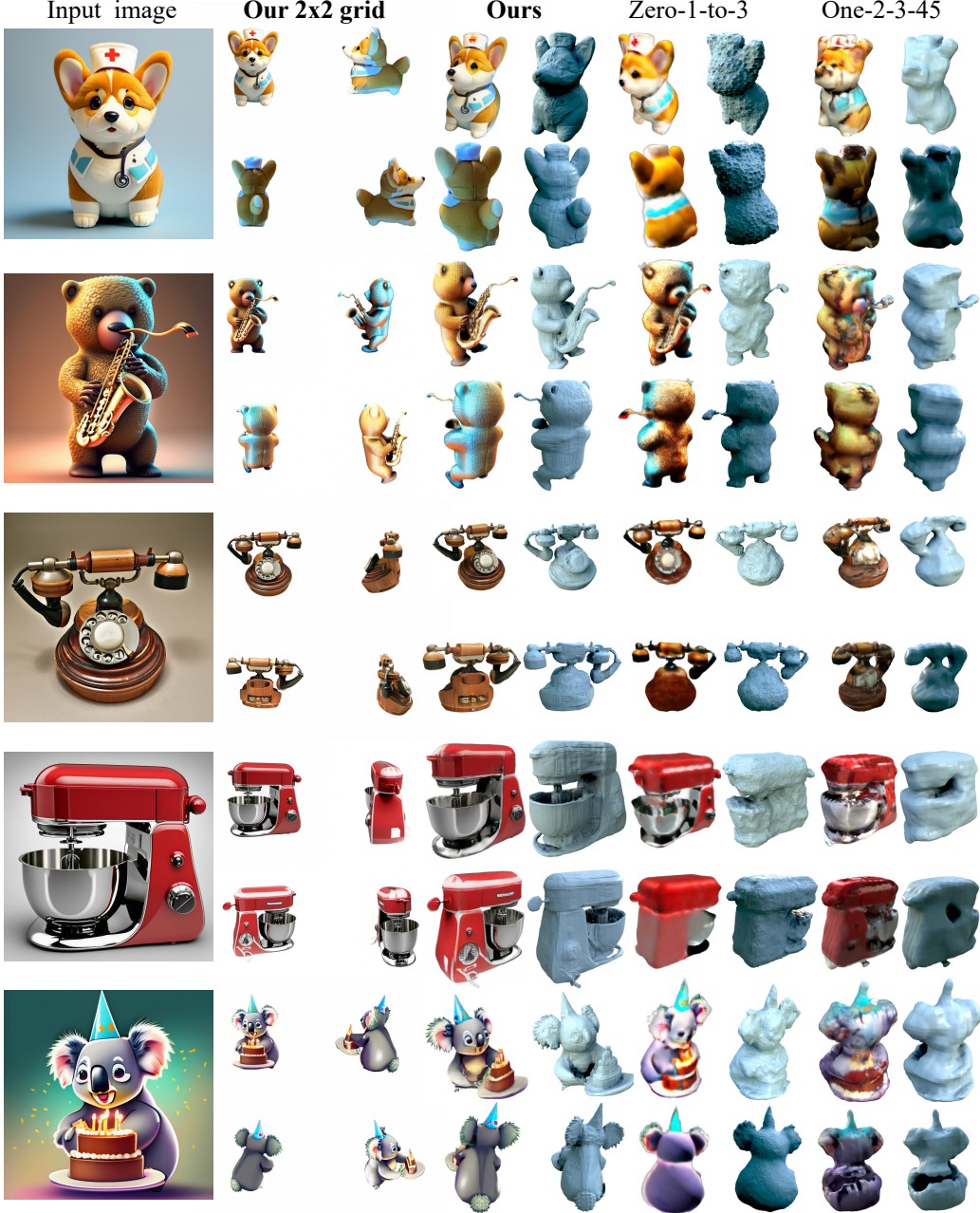

Figure 8: Comparison to previous methods on single image-conditioned 3D generation. We compared to previous methods Zero-1-to-3 (Liu et al., 2023b) and One-2-3-45 (Liu et al., 2023a). Our method can faithfully generate the details in the invisible regions, thus empowering us to reconstruct 3D assets of higher quality than baseline methods. All input images are generated with a public text-to-image platform Adobe Firefly (Adobe, 2023).

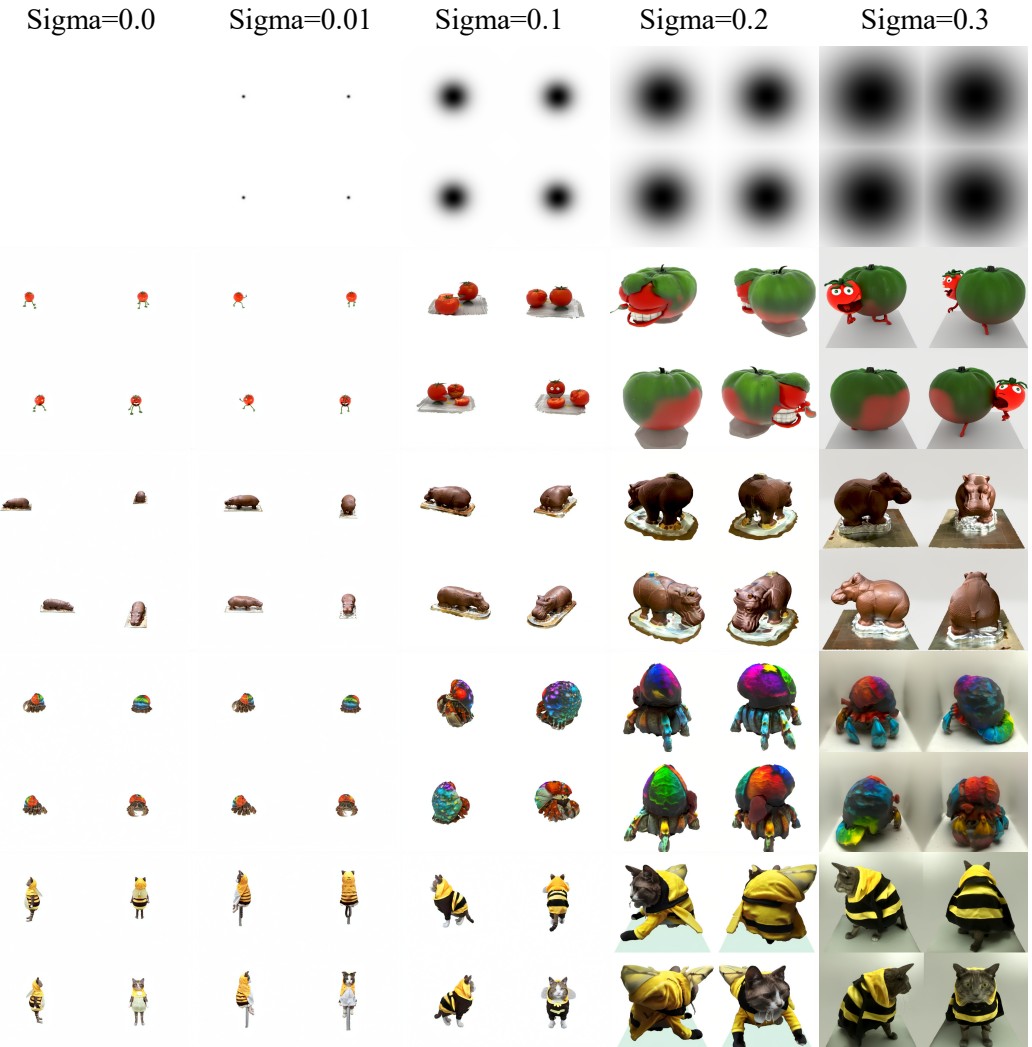

Figure 9: $2 \times 2$ grid images generated with Gaussian blobs of different sigma $\sigma$.

10 steps
(6 seconds)
20 steps
(8 seconds)
50 steps
(15 seconds)
100 steps
(20 seconds)

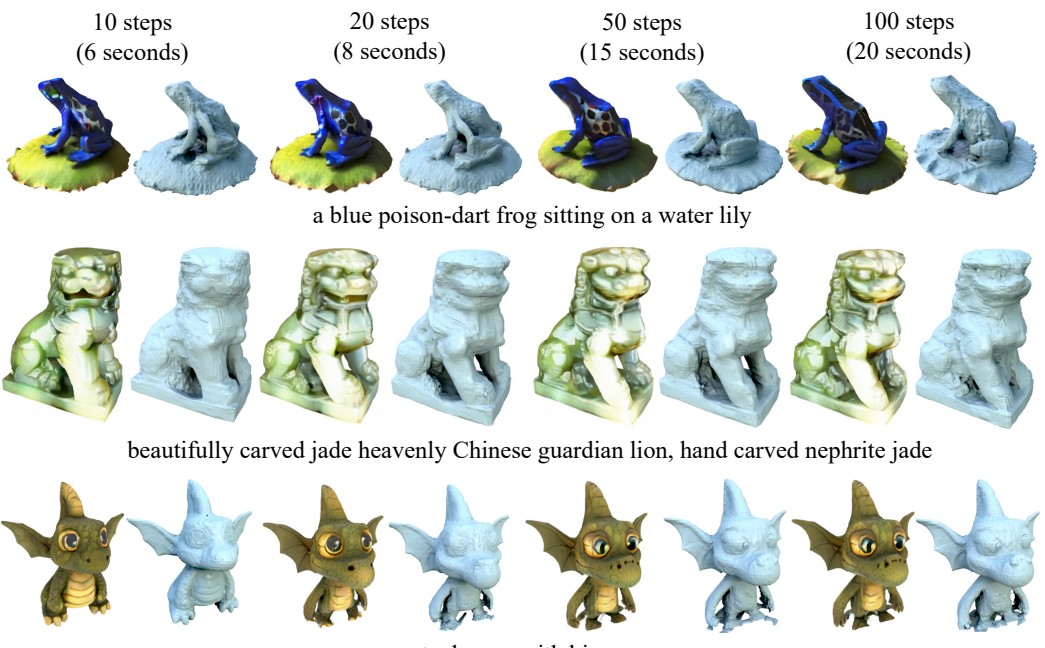

a blue poison-dart frog sitting on a water lily

beautifully carved jade heavenly Chinese guardian lion, hand carved nephrite jade

a cute dragon with big eyes

Figure 10: Comparison on the NeRF assets generated with different numbers of DDIM steps and their inference time. While we use 100 steps in our experiments that take 20 seconds to generate a NeRF asset, we find that using a smaller number of steps can also give us results of similar quality with a much shorter inference time.

**Ours**                                    ProlificDreamer

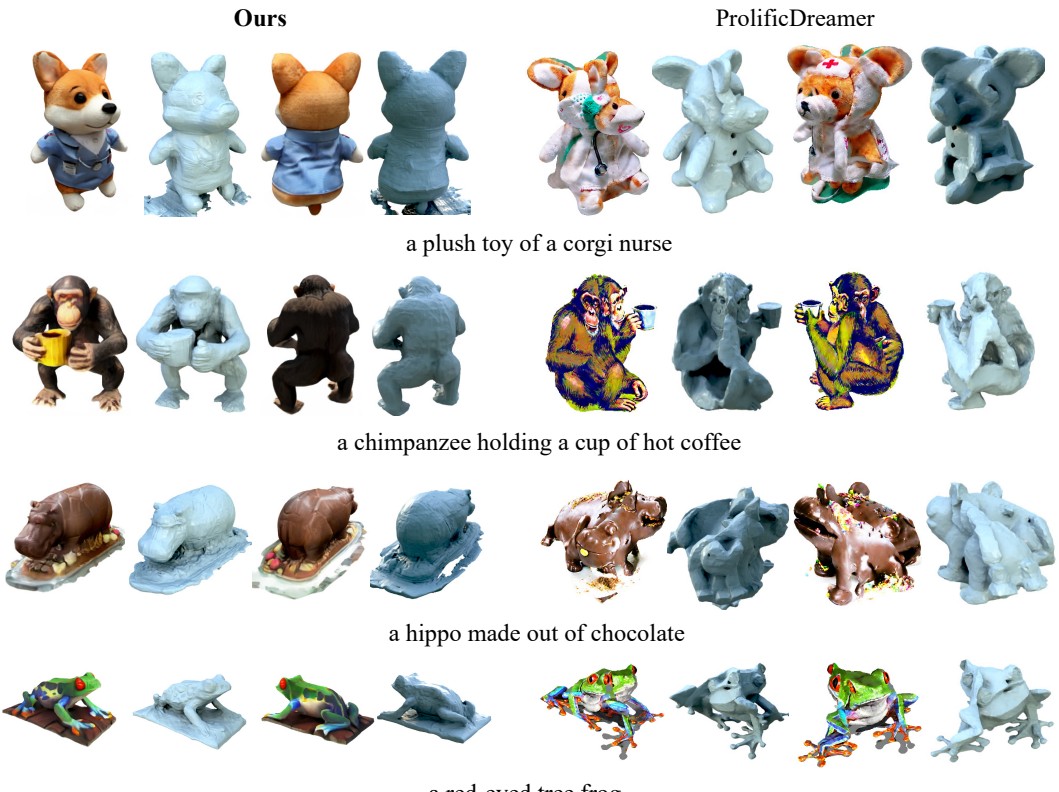

a plush toy of a corgi nurse

a chimpanzee holding a cup of hot coffee

a hippo made out of chocolate

a red-eyed tree frog

Figure 11: SDS optimization-based methods such as ProlificDreamer (Wang et al., 2023b) can possibly suffer from the Janus problem, which greatly degrades the quality of the 3D assets. In contrast, our method can mostly get rid of this problem.

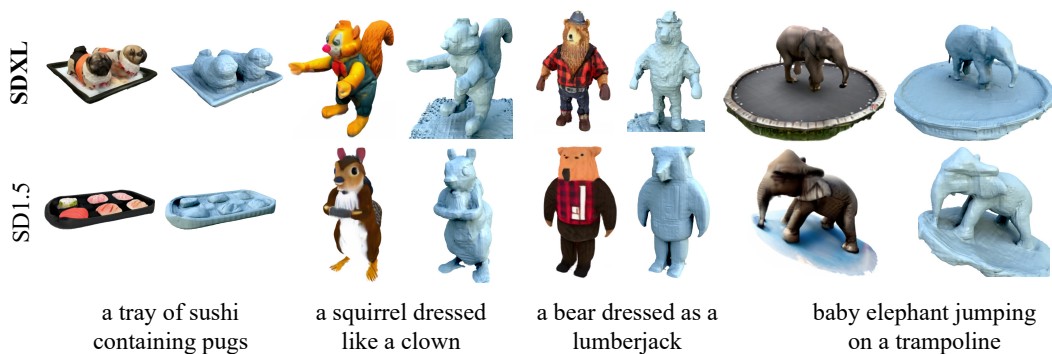

a tray of sushi          a squirrel dressed      a bear dressed as a       baby elephant jumping
containing pugs          like a clown            lumberjack                on a trampoline

Figure 12: Comparisons on the quality of the NeRF assets generated with fine-tuned SDXL and SD1.5 models. SDXL has a model size that is three times larger than SD1.5 and thus has better text comprehension. As shown in the figure, the 3D assets generated by our fine-tuned SDXL have better photo-realism and text alignment. The used SDXL and SD1.5 models are from Exp d (Curated-10K-s10K) and m (Curated-100K-s40K) in Table 3.

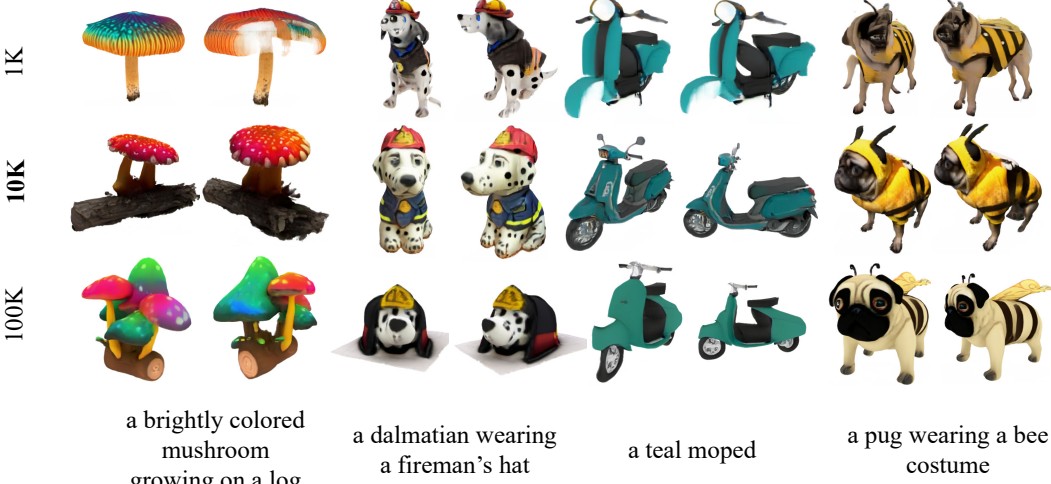

Figure 13: Comparison on the effect of different fine-tuning data sizes. Training on too little data such as 1K results in inconsistency between the generated 4 views, thus resulting in incorrect geometry. On the other side, training on too much data such as 100K makes the model biased toward the fine-tuning dataset, thus negatively affecting the quality of generated 3D assets. Here 1K, 10K and 100K correspond to Exp a (Curated-1K-s1K), d (Curated-10K-s10K) and g (Curated-100K-s40K) in Table 3 respectively.

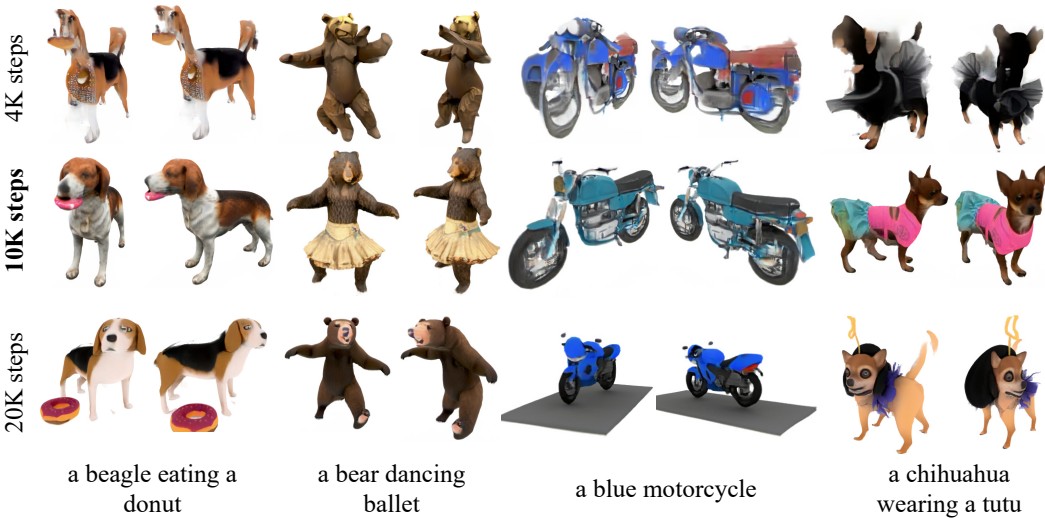

Figure 14: Comparison on different numbers of fine-tuning steps. 4K training steps lead to inconsistent 4-view generation, while 20K result in biasing towards the fine-tuning data. In contrast, 10K achieve a balance between these two. Here 4K, 10K and 20K correspond to Exp c (Curated-10K-s4K), d(Curated-10K-s10K) and e (Curated-10K-s20K) in Table 3.

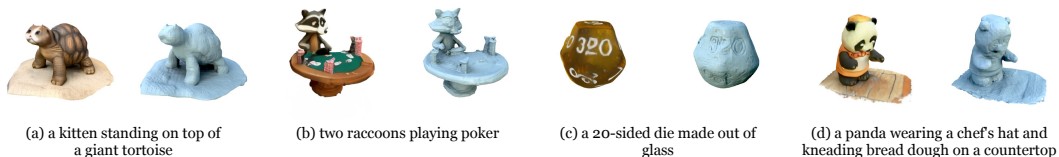

(a) a kitten standing on top of a giant tortoise    (b) two raccoons playing poker    (c) a 20-sided die made out of glass    (d) a panda wearing a chef's hat and kneading bread dough on a countertop

Figure 15: Some examples of our failure cases. (a) Incorrect understanding of compositional concepts. (b) Inability to generate the exact quantity. (c) Fail to generate objects with complex structures. (d) Missing important concepts in the prompt.

