# OpenReview forum: "Instant3D: Fast Text-to-3D with Sparse-view Generation and Large Reconstruction Model"
_ICLR.cc/2024/Conference — ICLR 2024 poster_

### Official Review · Reviewer_wyA8 · 2023-10-31

**Soundness:** 3 good
**Presentation:** 4 excellent
**Contribution:** 4 excellent
**Rating:** 8
**Confidence:** 5

**Summary:**

The paper presents a method for generating 3D shapes from text. The paper observes problems in the current score distillation-based optimization method, including slow inference, low diversity, and Janus problems, and proposes methods to solve them. As a first step, it proposed a text-conditioned sparse-view generation model, finetuned from a large-scale diffusion model of text-to-image generation. It is capable of generating high-quality sparse-view images without clustering the background. Second, it reconstructs the 3D shape based on the sparse views it generates. To reconstruct the sparse views, view-conditioned image tokens are encoded. Image tokens are concatenated from four views and fed into a triplane decoder. A NeRF decoder takes the decoded triplane features and reconstructs them into a 3D shape. Using the proposed method, high-quality and diverse 3D shapes can be created in 20 seconds. Using the proposed method, the Janus problem is prevented by maintaining shapes across views.

**Strengths:**

This paper is well written, clearly motivated, significantly contributed, and extensively experimented. The strengths I found about this paper include but are not limited to:
+ It proposes an effective method for resolving the Janus problem in text-to-image optimization-based method for text-to-shape generation. The experiment shows that the generated 3D shapes have better 3D structure and texture consistency across views.
+ It proposes a lightweight fine-tuning method and conducts extensive experiments for text-to-sparse view image generation. The method leverages the capability of a large-scale text-image generated model and proves that it has the ability to generate images across sparse views with light fine-tuning. I believe this model can not only contribute to text-to-shape generation but also to other domains.
+ It proposes an effective sparse-view reconstruction method that outperforms other sparse-view reconstruction methods in object-only datasets.
+ As a feedforward method, it generates 3D shapes efficiently within only 20 seconds.

**Weaknesses:**

The paper still has some limitations which I think are not discussed thoroughly:

+ Over-saturated problem. In Figure 4, the paper provides examples that have more photorealistic colors. However, it still suffers from an over-saturated problem to some extent, especially in the examples provided in Figure 5. I think the increment of texture quality majorly resulted from the curated training dataset, which removes cartoonish and low-quality instances, but not a result of improving the texture generate method itself(i.e. improving the rendering method, adding extra photo-realistic losses). If my interpretation is correct, I think this should be stated in the limitation section.
+ Resolution. As the author stated in the limitation section, generating four sparse view images leads to a degradation of texture quality. It would be better to provide a qualitative experiment by measuring the PSNR/SSIM/LPIPS of single image and multi-view images.
+ Diversity. In Figure 6, the paper provides examples showing the method is able to generate diverse results. My question is if the method provides more diverse results compared with other optimization-based methods. Will the feed-forward method be helpful in providing more diverse results than the optimization-based method practically? It would be better to provide some examples here.
+ In A.3 the paper detailed how to use CLIP features to filter out low-quality shapes. While I'm convinced the CLIP feature can filter out shapes with a cartoonish style, I'm not very convinced that the CLIP feature is able to tell apart shape quality. I hope the authors can provide some positive and negative examples here.
+ Some missing citations.
 1. Section 2.1 paragraph 1. Missing methods using implicit representation[1-3].
 2. Section 2.1 paragraph 2. Missing some diffusion-based generation methods[4].
[1] Towards Implicit Text-Guided 3D Shape Generation
[2] ShapeCrafter: A Recursive Text-Conditioned 3D Shape Generation Model
[3] CLIP-Forge: Towards Zero-Shot Text-to-Shape Generation
[4] CLIP-Sculptor: Zero-Shot Generation of High-Fidelity and Diverse Shapes from Natural Language

**Questions:**

+ Object-centric COCO. Considering all of the models are finetuned with the Objaverse-XL dataset, I'm wondering if it is still able to generate some shapes whose distribution is outside the Objaverse-XL dataset. I acknowledge it would be hard to prove, but I'm curious to see if the method is able to generate meaningful shapes in the Object-centric COCO dataset[1].
+ View condition. When training the view-conditioned image-to-triplane decoder, are the training shapes canonicalized or not? Let's say we input a set of views V = [v1, v2, v3, v4] and generate a shape A.  Then we multiply all the views with a transformation matrix M and generate a shape B. Will shape A and shape B under the same canonicalized coordinate frame?
+ Minor writing mistakes.
 1. Section 2.2. "unseeen" -> "unseen".
 2. Section A.4. "The dimension of each plane is 80 All three...." -> "The dimension of each plane is 80. All three...."

[1] DREAMFUSION: TEXT-TO-3D USING 2D DIFFUSION

---

> ### Author Response · Authors · 2023-11-17
>
> We thank the reviewer for appreciation of our technical contributions and high-quality results. We make replies to the questions from the reviewer as follows:
>
> * ***Color saturation***: To demonstrate our method, we have included more than 150 results in the paper/website (https://instant-3d.github.io/) and additional 300 results in the supplementary materials (see the instant3d_uncurated_comparison.pdf file). As far as we can tell, we do not observe very prominent color saturation problems when compared to the baseline methods such as DreamFusion and ProlificDreamer that consistently exhibit over-saturated colors. In Figure 7 of the paper, we also show results of our model fine-tuned on non-curated data randomly sampled from Objaverse. While the quality is worse than our model fine-tuned on curated data, it does not have over-saturation problems as those in the SDS-based methods. Therefore, we believe the main reason is that our method uses normal feed-forward diffusion inference (DDIM) while optimization-based methods use score distillation losses.
>
> * ***Resolution***: In the limitation section, we mentioned that the degradation of rendering quality is mainly due to limited resolution of the NeRF tri-plane predicted by our transformer-based reconstructor instead of the four view generation. It’s not fully clear to us what the reviewer means by “measuring the PSNR/SSIM/LPIPS of single image and multi-view images”, and we would be happy to make further replies if the reviewer can make further clarification.
>
> * ***Diversity***: Thanks for the suggestion. We update the anonymous website to include some comparisons on diversity against optimization-based methods DreamFusion and ProlificDreamer in https://instant-3d.github.io/rebuttal_diversity.html. As we can see, our method can generate more diverse results compared to the other two methods. For example, for the prompt “a chimpanzee dressed like a football player”, our method generates diverse players while DreamFusion and ProlificDreamer failed to do so.
>
> * ***Data filtering***: We update the anonymous website to include renderings of some of the positive and negative examples during data curation in https://instant-3d.github.io/rebuttal_curation.html. We also had included the IDs of the shapes in the objaverse dataset that we use to fine-tune the diffusion model in the openreview Supplementary Material (see the curated_data directory), which can be used to inspecting the objects on https://objaverse.allenai.org/ and also ensures the reproducibility.
>
> * ***Missing citations***: Thanks for pointing out! We will update the citations in the revised version.
>
> * ***Object-centric COCO***: We update the anonymous website to include some examples on object-centric COCO prompts: https://instant-3d.github.io/rebuttal_object_centric_coco.html. On the other hand, all the prompts we use are from the DreamFusion gallery (https://dreamfusion3d.github.io/gallery.html). These prompts are similar to Object centric COCO (e.g., it contains “a kitten looking at a goldfish in a bowl” whose style is like the COCO captions), and have complex compositional concepts (such as “a hippo biting through a watermelon”, “a pug wearing a bee costume”, “a squirrel dressed like Henry VIII King of England”) that does not exist in the Objaverse dataset. We believe that our results on these prompts show that our model can generalize well beyond the Objaverse dataset. The main text of the paper only shows a small amount of the results, please see the anonymous website https://instant-3d.github.io/ and the uncurated results in the openreview Supplementary Material (See the instant3d_uncurated_comparison.pdf file) for more results and their prompts. Note that we only perform a light-weight fine-tuning for only 10K steps, and therefore our fine-tuned model can still mostly preserve the capability and knowledge of the original SDXL model, which enables our model to generate shapes that are not in Objaverse from complex prompts.
>
> * ***View condition***: We do not canonicalize the shapes in Objaverse except that we scale all the objects to a cube within [-1, 1].  The input to the transformer includes both the images and their camera poses in the world coordinate system. Therefore, if we keep the images unchanged and only transform the camera poses, the output object will be transformed correspondingly.
>
> * ***Minor mistakes***: Thanks for pointing them out! We will correct them in our revised version.

---

> > ### Author Response · Authors · 2023-11-21
> >
> > Dear reviewer, thanks again for your comments and suggestions! Please kindly let use know whether our response has properly addressed your concerns and questions. We are happy to answer any other questions you may have. We would also appreciate your response regarding whether you might be willing to raise your score. Thank you!

---

> > ### Comment · Reviewer_wyA8 · 2023-11-22
> >
> > Thanks to the authors for your feedback and for providing more results.
> > + Resolution. I misunderstood the cause of the downgrading of the image quality, and the author resolved my question.
> > + Missing citation. One extra paper for the implicit-based generation I missed. See [1].
> > [1] AutoSDF: Shape Priors for 3D Completion, Reconstruction and Generation
> >
> > In general, this is a good paper and proposes a lightweight model for resolving the multi-view correspondence problem in 3D generation. One thing I feel is a little under-interpreted is the canonicalization of the objects. The current viewpoints are under the world coordinate systems, but the Objaverse shapes are un-canonicalized. So, how does the coordinate system interpret the coordinates of a generated object? Would the coordinate system resemble the object in the database that is most similar to the generated object? Nonetheless, I think this is a good paper for an ICLR submission.

---

> > > ### Author Response · Authors · 2023-11-23
> > >
> > > We thank the reviewer for the comments.
> > >
> > > In terms of canonicalization, one thing that’s worth mentioning is that the 3D objects in Objaverse already have a consistent up direction (with z up), this provides some kind of canonicalization that facilitates the generation. But we would like to point out that the objects’ orientations around the z up vector have a lot more variations and we don’t do any canonicalization. When trained on such data, both our fine-tuned SDXL model and our reconstruction model are able to handle these variations. In particular, we note that our fine-tuned SDXL generates 4 views with adjacent views’ azimuth difference being 90 degrees, as a result of our view selection scheme in our training data during fine-tuning. By providing our reconstructor with a set of 4 poses (not unique) that respect this 90-degree azimuth difference constraint, our reconstructor is able to reconstruct a plausible 3D NeRF from the generated images (please refer to https://instant-3d.github.io/rebuttal_coordinate.html for an example. We rotate the camera poses by 90 degrees each step and use the rotated poses as input to reconstruct the shape from the original 4-view images. As we can see from the result, the reconstructed shape will be rotated correspondingly).

---

### Official Review · Reviewer_eZrX · 2023-11-01

**Soundness:** 3 good
**Presentation:** 3 good
**Contribution:** 3 good
**Rating:** 8
**Confidence:** 5

**Summary:**

The authors propose a framework for text-to-3D generation in a feed-forward manner, without requiring an optimization loop during inference. The approach first generates multi view images from a text prompt and gaussian blob initialization. The multiview images are then fed through a transformer based reconstruction network that generates a triplane, which can then be used for volume rendering novel views. State of the art performance is  demonstrated compared to recent text-to-image baselines.

**Strengths:**

1. **Novelty**: The ideas introduced in this manuscript are reasonably novel. In particular, gaussian blob initialization for multiview generation is a potentially useful trick that can be applied to a variety of text-to-3D or image-to-3D pipelines.
1. **Paper quality**: The paper is well-written and clearly presented, with attention to detail. The authors have clearly put a lot of effort into making the paper easy to read and understand.
3. **Related work**: An adequate treatment of related works have been provided to place this work in the context of current literature.
4. **Reproducibility**: The exact details of the approach, architecture specifics and training details have been provided to aid in the reproducibility of the approach. Furthermore, finetuning datatset information has also been provided in the supplm.
2. **Comparisons**: The paper provides adequate comparisons to baselines, which is important for demonstrating the effectiveness of the proposed approach. Implementations of Dreamfusion on IF has been used as a strong baseline
3. **Ablation**: Ablation studies are provided to highlight the need for each of the components introduced. Particularly, the motivation for the gaussian blob initialization and finetuning on different data.
4. **Approach**: The proposed solution of generating 4 views is interesting and adds to the multiview consistency to some extent.
5. **Appendix**: The authors provide a clear and detailed appendix section with additional reference to LRM for Image-to-3D reconstruction. A number of uncurated text to image examples are provided.

**Weaknesses:**

1. **Need for gaussian blob**: How important is it for the initialization to be a gaussian blob? Can’t the same effect be achieved with a square mask since the primary intent is to localize the generated outputs to a region?
3. **Image features**: How important are the Dino features? In particular, is there a significant drop in performance with features obtained from other pre-trained networks? Ablation with say VGG or other conv features would be insightful to determine the importance of the choice of features.
5. **Comparison**: Additional comparison to amortized text-to-3D approaches like ATT3D[1] both in terms of quality and in terms of compute and inference costs will help highlight the contributions of this work. Additionally, most of the comparisons are against volume synthesis methods, how does the quality compare to mesh synthesis methods like Magic3D[2] ?
6. **Novel view consistency**: It is unclear how multi-view consistent the rendered novel views are. Although table 2 provides comparison of pixel aligned metrics against SparseNeus, the work would greatly benefit by presenting video results of turntables of the rendered objects. This will help with the qualitative evaluation of the multiview consistency of the object.
7. **Tiled generation vs multichannel**: Although contemporary to this work, motivating the need for tiling the views as opposed to generating them as separate channels as in MVDream[3]. Strict qualitative comparisons are not warranted, but highlighting the advantage of the tiled 4 view representation (particularly, since this reduces the resolution) would be insightful.
8. **Number of view**: Section 3.1 mentions trade-off of number of views vs quality. Providing some qualitative/ quantitative justification for this (either in the appendix or supplm) would be very helpful.
9. **Data distribution**: Since stage 2 is only trained on Objaverse-XL renders, is there an issue with the kinds of 3D assets that can be generated? In particular, the generated assets look synthetic and from the distribution of Objaverse instances.
10. **Choice of Diffusion model**: The tiled approach works well for latent space models like SD and SD-XL due to the inherent high resolution input output. Can this framework also be adapted in pixel space diffusion models like DeepFloyd. Providing some insight for this will be helpful in determining the choice of diffusion model.
10. **SD vs SDXL**: Although quantitative evaluations are presented, providing some qualitative comparison of assets generated from SD finetuning vs SDXL finetuning would be helpful (in appendix or supplm).

[1] Lorraine et al. ATT3D, ICCV23.
[2] Lin et al. Magic3D, CVPR23.
[3] Shi et al. MVDream arxiv23

**Questions:**

1. How important are DINO features?
2. What is the advantages of tiled generation of 4 views over generating on multiple channels.?

---

> ### Author Response · Authors · 2023-11-17
>
> We thank the reviewer for the acknowledgement of the technical novelty and state-of-the-art performance of our methods. We make replies to the questions from the reviewer as follows:
>
> 1. ***Gaussian blob***: Following the reviewer’s suggestions, we try replacing the Gaussian blobs with square bounding boxes at the center, and update the anonymous website with some example results (https://instant-3d.github.io/rebuttal_blobs.html). As we can see from the results, the square bounding boxes work reasonably well when they are relatively small, but suffer from obvious degradation in image quality and introduce undesired black background when the size increases.
>
> 2. ***Image features***: We provide our thoughts from three perspectives and would like to share some empirical observations.
>
>     **(a) Pre-trained vs from-scratch**. We have trained our model without initializing the visual encoder from a pre-trained weight (i.e., trained it from scratch). We found that a pre-trained model gives faster convergence in the beginning, but the gap keeps decreasing as the training budget increases. We have not seen the complete elimination of the gap under our limited compute budget but we guess that the gap can possibly disappear with more data and longer training time.
>
>     **(b) Different pre-trained methods**. For the same ViT architecture, there are multiple pre-trained methods, e.g., MAE, DINO, CLIP, and ImageNet-CLS. Among them, we have tested with CLIP-init visual encoder but the results are worse than DINO. We guess that it is because CLIP is only trained for its [CLS] token with img-text contrastive loss thus it is hard to preserve all raw spatial information. For the other two (i.e., MAE and IN-CLS), we have not given a test. But we guess that MAE might perform similar to DINO, and IN-CLS would perform similar to CLIP based on how their pre-training method is built: MAE supervises spatial tokens while IN-CLS only supervises the [CLS] token. DINO supervises [CLS] token as well but its spatial token is good according to its pre-training strategy and also shown in previous papers [1].
>
>     **(c) Different model architectures (e.g., ViT vs CNN)**. We have not experimented with CNN-based visual encoder, but we do think that CNN should give similar results. The only consideration in our mind is that CNN usually needs a different training schedule than transformer thus makes the model tuning harder. Given this, we pick ViT in our exploration since easy tuning is crucial for large-scale training.
>
> 3. ***Comparison***: We haven’t been able to compare to ATT3D because there is no public implementation. As for mesh synthesis methods, we compare to ProlificDreamer (please see Figure 4 in the paper and the uncurated comparison in the supplementary materials)  that uses a three-stage optimization method (NeRF -> texture refinement -> mesh refinement) to generate meshes with textures, similar to Magic3D.
>
> 4. ***Novel view consistency***: We would like to point out that our novel views are rendered from a NeRF in the format of a triplane (predicted by our transformer-based reconstructor). Therefore, it’s view-consistent by the nature of volume rendering. We provide turntable videos of our generated objects in the anonymous website https://instant-3d.github.io/ where we can see that the rendered views are consistent with each other.
>
> 5. ***Tiled generation vs Multi-Channel***: One main advantage of our tile-based generation is it does not need to change the architecture of the original diffusion model. Multi-channel methods rely on specific attention modules in the original diffusion architecture, and need to add extra parameters such as view conditioning to help the model distinguish between different views. In contrast, tile-based generation can be applied to a wide range of network architectures without modifying the network architectures or any extra inputs. Therefore, our tile-based method has several benefits like:
>
>     **(a)** The method is more transferable among different generative methods and model architectures without specific adaptation.
>
>     **(b)** The light-weight fine-tuning can be easily applied to the new model.
>
>     **(c)** Any acceleration techniques developed for the base model can be directly applied, thus our method can easily enjoy the fastest deployment developed by the community.
>
>     **(d)** Similar to (c), any on-device/platform-specific code (e.g., cpp code; swift code) can be easily applied with our model as well, which enables more potential applications.
>
>     In the meanwhile, we agree with the reviewer that multi-channel generation is an interesting future direction to explore.
>
> ***(1/2)***

---

> > ### Author Response · Authors · 2023-11-17
> >
> > 6. ***Number of views***: Thanks for the suggestion. Carefully validating the performance of different numbers of views quantitatively is interesting but would require extra computing resources and time, and we hope to present it when possible.
> >
> > 7. ***Data distribution***: We would like to clarify that the second-stage model is trained on Ojaverse not Objaverse-XL. Also we want to point out that the second-stage sparse view reconstructor model is a fully deterministic model with no generative capability. It learns to reconstruct the NeRF by using information from the generated multi-view inputs from the first stage, instead of “guessing” what the object would look like. We update the anonymous website to include the generated four-view images and our reconstructed NeRFs https://instant-3d.github.io/rebuttal_image2nerf.html. From the results, we can see that our reconstructor can faithfully reconstruct the appearance of the objects that closely match the generated four views, which demonstrates the robustness and generalization of our reconstructors. For the first stage, we have conducted a light-weight fine-tuning for only 10K steps to preserve the capability of the SDXL model to the most extent, and therefore it can generate 3D objects from complex compositional prompts that are out of the distribution of the Objaverse dataset (such as “a hippo biting through a watermelon”, “a pug wearing a bee costume”, “a squirrel dressed like Henry VIII King of England”). That being said, we do observe that there still exists a gap in photorealism between our fine-tuned model in the first stage and that of the original SDXL model, which may be caused by the discrepancy in training data distribution. We leave it as future works to further bridge such a gap.
> >
> > 8. ***Choice of diffusion model***: The advantage of using latent diffusion model like SD and SDXL is that it can generate high resolution images in one shot, while models like DeepFloyd involves multiple upsampling stages, which introduce extra complexity when we want to fine-tune it for generating view consistent images.
> >
> > 9. ***SD vs SDXL***: We refer the reviewer to Figure 12 for visual comparisons between these two models.
> >
> >
> > [1] Deep vit features as dense visual descriptors, Amir, S., Gandelsman, Y., Bagon, S., & Dekel, T. (2021).
> >
> > ***(2/2)***

---

> > > ### Author Response · Authors · 2023-11-21
> > >
> > > Dear reviewer, thanks again for your comments and suggestions! Please kindly let use know whether our response has properly addressed your concerns and questions. We are happy to answer any other questions you may have. We would also appreciate your response regarding whether you might be willing to raise your score. Thank you!

---

> > > > ### Comment · Reviewer_eZrX · 2023-11-23
> > > > **Response to queries**
> > > >
> > > > The authors do a great job at addressing all the concerns raised. Additionally, with the text-to-3D and image-to-3D problems gaining popularity of late, this work would serve as a strong baseline for future work in this space. To that end I keep the current score and recommend the paper for acceptance.

---

### Official Review · Reviewer_dCt6 · 2023-11-01

**Soundness:** 3 good
**Presentation:** 3 good
**Contribution:** 2 fair
**Rating:** 6
**Confidence:** 3

**Summary:**

The paper proposes a 3D distillation method from fine-tuned text-to-2D diffusion models fintuned. The method tackles the diversity and Janus problem in prior methods and achieves significant speedup compared to prior approaches.

**Strengths:**

* There are several technical components proposed in the method to achieve good visual quality.
* The speedup compared to prior optimization-based methods is significant.

**Weaknesses:**

* In Figure 11, the paper claims to get rid of the Janus problem, but such a claim should be rigorously verified across a large set of text prompts instead of using the selected examples.
* The paper proposes to use a feedforward transformer for sparse-view reconstruction, and both this model and the fine-tuned Stable-Diffusion model are trained on the Objaverse dataset, which can potentially introduce a large domain gap when applying the model to arbitrary text prompts. A discussion on failure cases related to the domain gap, if there are prominent ones, may help readers better assess its applicability.

**Questions:**

* The paper provides qualitative examples suggesting an improved diversity compared to prior methods but lacks a discussion on which technical component in the proposed pipeline contributes to such diversity.

___
Post-rebuttal response:
I've read comments from other authors and the rebuttal responses.
* I am convinced that the output 3D asset quality from this work and the speedup compared to prior works are significant based on the thorough experiments and examples shown in the paper.
* The performance gain largely rely on the the powerful backbone model, which shares a lot of similar design choices and training strategies compared to LRM which is appended in the supplementary and briefly discussed, but not directly compared to, in the paper.

I'm raising my score to be 6 based on the performance and additional failure cases analysis during the rebuttal. Additional comparisons to a simple baseline adapting LRM could further help clarify the contribution delta of this work.

---

> ### Author Response · Authors · 2023-11-17
>
> We thank the reviewer for appreciation of the visual quality and efficiency of our method. We make replies to the questions from the reviewer as follows:
>
> * ***Janus problems***: We include more than 150 results from a diverse set of prompts in the paper/website (https://instant-3d.github.io/) and additional 300 uncurated comparison results in the openreview Supplementary Material (see the instant3d_uncurated_comparison.pdf file). As we can see from the results, a vast majority of them are free from the Janus problem.  We are happy to tone the claim down and submit a revised version before the rebuttal ends.
>
> * ***Domain gap***: While our first-stage sparse view generation model is fine-tuned on objaverse dataset, we would like to point out that we have conducted a light-weight fine-tuning for only 10K steps, which enables the model to mostly preserve the original capability of the SDXL model to handle a wide range of prompts including complex compositional concepts that are not available in the objaverse dataset, as demonstrated in our paper, website (https://instant-3d.github.io/, e.g., “a squirrel dressed like Henry VIII King of England”), and openreview Supplementary Materials (i.e., instant3d_uncurated_comparison.pdf). In the meanwhile, our second-stage sparse-view reconstruction is a deterministic model that takes multi-view images from the first-stage as input and lift them to 3D, which is trained on renderings from Objaverse dataset that cover a wide range of different shapes and allows the model to learn generic 3D priors for reconstruction. We observe that the model can work well on generated images and didn’t notice obvious failures in this stage.  We update the anonymous website to include the generated four-view images and our reconstructed NeRFs https://instant-3d.github.io/rebuttal_image2nerf.html. From the results, we can see that our reconstructor can faithfully reconstruct the appearance of the objects that closely match the generated four views, which demonstrates the robustness and generalization of our reconstructors.
>
>     That being said, we do observe that our model fails to handle some over-complicated prompts, for example, those related to complex spatial arrangements of multiple subjects and complex scenes. In addition, the generated assets also are not as photorealistic as the 2D images generated by SDXL, which may be attributed to the information loss in the fine-tuning stage due to domain gaps. One possible solution to this problem is to include single-view natural images such as those from LAION dataset in the fine-tuning, and we believe this will be an interesting future work. We will include more examples of failure cases and discussions in the revised version.
>
> * ***Diversity***: Our text-to-multiview generation network is fine-tuned from a 2D text-to-image diffusion model SDXL, and we apply regular diffusion sampling and denoising methods during inference. Therefore, it naturally inherits the capability of the 2D diffusion models in generating diverse images from random samples, which are then lifted to 3D to generate diverse 3D assets using our sparse-view reconstructor. In contrast, previous methods such as DreamFusion and ProlificDreamer apply Score Distillation-based optimization, which are prone to generate similar results even with different initializations, as discussed in Section A.5 of the DreamFusion paper [1]. We also updated our anonymous website to include some comparisons on diversity with DreamFusion and ProlificDreamer, please check it via https://instant-3d.github.io/rebuttal_diversity.html where we generate 4 different shapes for each prompt with different seeds. From the results we can observe that our method generates more diverse results than baseline methods. For example, for the prompt “a chimpanzee dressed like a football player”, our method generates diverse players while DreamFusion and ProlificDreamer failed to do so.
>
> [1] Dreamfusion: Text-to-3D using 2D Diffusion, Poole et. al, https://arxiv.org/abs/2209.14988 (2022)

---

> > ### Author Response · Authors · 2023-11-21
> >
> > Dear reviewer, thanks again for your comments and suggestions! Please kindly let use know whether our response has properly addressed your concerns and questions. We are happy to answer any other questions you may have. We would also appreciate your response regarding whether you might be willing to raise your score. Thank you!

---

> > ### Author Response · Authors · 2023-11-22
> > **Reminder for the review update**
> >
> > Dear reviewer, just another kind reminder before the official rebuttal period ends.
> >
> > We have provided additional results for the concern of `Reconstruction domain gap` (https://instant-3d.github.io/rebuttal_image2nerf.html) and `Generation diversity` (https://instant-3d.github.io/rebuttal_diversity.html) here. We also refer to 150 + 300 diverse results in our paper/website (https://instant-3d.github.io/) and Supplementary Material (instant3d_uncurated_comparison.pdf file) to support our `reducing Janus problem` claim.
> >
> > As we think that these materials can resolve reviewer's all three concerns, we kindly request the reviewer to have a look and also reflect the feedback in the official review. If there are more concerns, we are happy to resolve them.

---

### Meta-Review · Area_Chair_iDZM · 2023-12-07

**Metareview:**

This work proposes a technique to generate multi-view images from a pretrained generative model followed by using a large transformer based network to predict 3D triplanes. All the reviewers lean towards accepting the work. Reviewers appreciated the high-quality results. Some reviewers raised concerns related to relative merits of this work in related to LRM and PF-LRM works. After the extensive discussions, it is felt that this is work has enough contributions to be accepted. The reviewers did raise some valuable concerns that should be addressed in the final camera-ready version of the paper, which include adding the relevant rebuttal discussions and revisions in the main paper. The authors are encouraged to make the necessary changes to the best of their ability.

**Justification For Why Not Higher Score:**

Minor contributions w.r.t concurrent submissions from the same authors.

**Justification For Why Not Lower Score:**

All reviewers are leaning towards acceptance.

---

### Decision · Program_Chairs · 2024-01-16

Accept (poster)